# LONGITUDINAL LATENT DIFFUSION MODELS

## ABSTRACT

Longitudinal data are crucial in several fields, but collecting them is a challenging process, often hindered by concerns such as individual privacy. Extrapolating in time initial trajectories or generating fully synthetic sequences could address these issues and prove valuable in clinical trials, drug design, and even public policy evaluation. We propose a generative statistical model for longitudinal data that links the temporal dependence of a sequence to a latent diffusion model and leverages the geometry of the autoencoder latent space. This versatile method can be used for several tasks - prediction, generation, oversampling - effectively handling high-dimensional data such as images and irregularly-measured sequences, needing only relatively few training samples. Thanks to its ability to generate sequences with *controlled* variability, it outperforms previously proposed methods on datasets of varying complexity, while remaining interpretable.

## 1 INTRODUCTION

Longitudinal data, also known as panel data, consist of repeated measurements over time that track the evolution of the same entity or individual—more concretely, its *trajectory*. The total number of observations is relatively small, and their frequency can be sparse, unlike time series data, which typically involve more frequent measurements. Longitudinal data are common in many application fields, such as medicine (e.g., for modeling disease progression (Zhao et al., 2021) or monitoring treatment response (Blackledge et al., 2014)) and econometrics (Baltagi, 1995).

Their dimensionality can range from relatively low (e.g. tabular data) to quite high (e.g. images). Furthermore, the number of different entities followed is most of the time pretty small (in the case of rare diseases for instance). These aspects make them challenging to model; still, the generation of synthetic longitudinal data can have powerful applications (Mosquera et al., 2023; Kühnel et al., 2023), for data augmentation, future prediction and missing data imputation.

In our applications, the ideal generative model for longitudinal data needs to produce varied trajectories while starting from the same situation, but this variability needs to be controlled to limit variations *around* one or several core tendencies [1], so that the generated samples remain plausible.

RELATED WORK

**Modeling longitudinal data**   Prior methods to statistically model longitudinal data and understand underlying evolution dynamics were mainly relying on mixed-effect models (Laird & Ware, 1982) that parameterise a patient's evolution as a deviation from a reference trajectory (Diggle et al., 2002; Singer & Willett, 2003; Debavelaere et al., 2020). These methods are quickly limited and can not be applied to complex trajectories (especially when there is no clear average evolution). Other modelling tentatives include RNN-based (Cao et al., 2018) or GAN-based methods (Luo et al., 2018; Sun et al., 2021). The former is not generative - only handling missing data, and the latter relies on a difficult adversarial training and does not yield a tractable and interpretable mathematical model.

**Improving VAE latent space**   Variational Autoencoders (VAEs) (Kingma & Welling, 2014; Rezende et al., 2014) are powerful models for capturing distributions. Their latent spaces can reveal representative features through disentanglement (Higgins et al., 2017) and can be equipped with

---

[1]Simplistically, think of a child's growth curve. Their height at a given age *conditions* the future trajectory, yet does not completely determine it. Generated plausible trajectories need to cater to this issue.

Riemannian geometry (Shao et al., 2018) to extract population structure in latent space. However, standard VAEs, assuming i.i.d. representations, fail to capture temporal correlations in data. Several works have aimed to enhance latent representations using Gaussian processes (Fortuin et al., 2020; Ramchandran et al., 2021) or normalizing flows (Rezende & Mohamed, 2016; Chadebec & Allassonnière, 2023). Yet, these models are mainly designed for missing data imputation or conditional tasks, making them less suitable for unconditional sequence generation. Moreover, normalizing flows, being deterministic, fail to introduce variation. Closely related to our work, Li & Mandt (2018) propose disentangling time-dependent features by jointly training a VAE with LSTMs, which results in a complex and computationally heavy training process.

**ODE / SDE**    Also closely related to our method are approaches involving neural ordinary differential equations (NODE) that see the forward pass of a residual network as solving an ODE. In particular, the latent neural ODE model proposed by Chen et al. (2018) defines a generative model by assuming that the initial state latent variable follows a given prior distribution and a latent trajectory is then obtained by solving an ODE. Yildiz et al. (2019), Kanaa et al. (2021) extends this method for high-dimensional data but unlike ours, these models are completely deterministic in the latent space, hindering the diversity of generated samples. Going stochastic, Li et al. (2020) use a latent SDE model but do not apply it to high-dimensional datasets.

**High-dimensional data generation**    Finally, our work relates to image generation method. Particularly, diffusion models - relying on SDE knowledge - have been crucial into generating high-resolution samples (Sohl-Dickstein et al., 2015; Ho et al., 2020). Rombach et al. (2022) used the latent space of a pre-trained VAE to even improve the quality and speed of training. Now, these techniques are also used for video generation (Ho et al., 2022; Lu et al., 2024) making them closer to longitudinal data. However, video and longitudinal image generation stay very different by essence, with regards to the frequency of images (very large in video, low and irregular in longitudinal), the necessary interpretability in the longitudinal case and more importantly, the needed number of training samples (huge for video generation, necessarily low for longitudinal data).

### OUR CONTRIBUTION

We propose here a new generative model for longitudinal data that uses a latent diffusion to model the time dependency between the observations of a given sequence: each embedding of the observation of a given sequence is forced to lie on a diffusion trajectory in a VAE latent space.

We demonstrate that our proposed method, named the Longitudinal Latent Diffusion Model (LLDM), excels at unconditionally generating fully synthetic trajectories with high performance, as well as generating trajectories conditioned on one or more input observations. A key strength of LLDM lies in the diversity of the sampled sequences, even when it is done conditionally, which is made possible by leveraging the inherent stochastic nature of diffusion processes.

## 2    BACKGROUND

### 2.1    VARIATIONAL INFERENCE AND A GEOMETRIC PERSPECTIVE ON VAES

Let $\boldsymbol{x} = (\boldsymbol{x}_i)_{1 \leq i \leq n} \in (\mathbb{R}^D)^n$ be a training dataset, i.i.d. from an unknown distribution $p(\boldsymbol{x})$. A variational autoencoder (VAE) is a generative model that aims at approximating $p(\boldsymbol{x})$ by a distribution on $\mathbb{R}^D$ parametrized with a neural network: $p_\theta$ with $\theta \in \Theta$. Ideally, the training of a VAE aims at minimizing the Kullback-Leibler divergence between $p$ and $p_\theta$, that is solving $\min_{\theta \in \Theta} D_{\mathrm{KL}}(p\|p_\theta)$. This is exactly equivalent as maximizing the joint log-likelihood: $\max_{\theta \in \Theta} \mathbb{E}_{y \sim p}[\log p_\theta(y)] \approx \log p_\theta(\boldsymbol{x}) = \frac{1}{n} \sum p_\theta(\boldsymbol{x}_i)$.

A VAE relies on variational inference, assuming that the generation process involves latent variables $\boldsymbol{z} \in \mathbb{R}^d$ with $d << D$. These variables have an assumed prior (let it be here a standard normal) distribution $\boldsymbol{z} \sim p(\boldsymbol{z})$ and then $\boldsymbol{x} \sim p_\theta(\boldsymbol{x}|\boldsymbol{z})$ is assumed to be a simple distribution, parametrized as a neural network known as the *decoder* (let it be a diagonal Gaussian, $\mathcal{N}(\mu_\theta(\boldsymbol{z}), \mathrm{diag}(\sigma_\theta^2(\boldsymbol{z})))$. Unfortunately, despite the simplicity of these distributions, the joint log-likelihood $p_\theta(\boldsymbol{x})$ (see Equation 1) and the posterior $p_\theta(\boldsymbol{z}|\boldsymbol{x})$ are intractable. The latter is approximated using a variational distribution $q_\phi(\boldsymbol{z}|\boldsymbol{x})$, parametrized as a neural network known as the *encoder* (here, $\mathcal{N}(\mu_\phi(\boldsymbol{x}), \mathrm{diag}(\sigma_\phi^2(\boldsymbol{x})))$).

Eventually, the VAE is trained minimizing the ELBO objective $\mathcal{L}$, defined as follow:

$$\log p_\theta(\boldsymbol{x}) = \log \int_{\mathcal{Z}} p_\theta(\boldsymbol{x}|\boldsymbol{z})p(\boldsymbol{z})d\boldsymbol{z} \geq \mathbb{E}_{z \sim q_\phi(\cdot|\boldsymbol{x})}[\log p_\theta(\boldsymbol{x}|\mathrm{z})] - D_{\mathrm{KL}}(q_\phi(\boldsymbol{z}|\boldsymbol{x})\|p(\boldsymbol{z})) . \quad (1)$$

With diagonal Gaussians $q_\phi$ and $p_\theta$, the first part of the ELBO is a *reconstruction loss*, while the second part (the Kullback-Leibler divergence) is a *regularization loss*.

A trained VAE can easily generate new data points by sampling $z$ from the prior distribution $\mathcal{N}(0, \mathbf{I})$ and decoding it using $p_{\theta^*}(\cdot|\boldsymbol{z})$. Yet, Chadebec & Allassonniere (2022) showed that its latent space can be seen as a Riemannian manifold $\mathcal{M} = (\mathbb{R}^d, \mathbf{G})$ where $\mathbf{G}$ is a smooth continuous Riemannian metric defined on $\Omega \subset \mathbb{R}^d$ that has a closed form. Using a Hamiltonian Monte-Carlo (HMC) sampler (see details in Appendix D), it is thus easy to sample $\boldsymbol{z}$ from the Riemannian uniform distribution on $\mathcal{M}$, defined as:

$$\mathcal{U}_{\mathrm{Riem}}(\boldsymbol{z}; \mathcal{M}) = \frac{\sqrt{\det \mathbf{G}(\boldsymbol{z})}}{\int_{\Omega} \sqrt{\det \mathbf{G}(z)dz}} . \quad (2)$$

This *geometry-aware sampling* leads to higher-quality generated samples as it enables to explore more the latent space than a blind standard normal sampling.

## 2.2 Latent diffusion models

Denoising Diffusion Probabilistic Models (DDPM) (Ho et al., 2020) are latent variable models designed to learn a data distribution $p$ by gradually denoising a normally distributed variable. It does so by learning the reverse process of a fixed Markov chain, called the *forward diffusion process* that gradually adds Gaussian noise to the data $\mathbf{x}_0 \sim q(\mathbf{x}_0)$, following a variance schedule $\beta_1, ..., \beta_{T_{\mathrm{diff}}}$:

$$q(\mathbf{x}_{1:T_{\mathrm{diff}}} \mid \mathbf{x}_0) := \prod_{t=1}^{T_{\mathrm{diff}}} q(\mathbf{x}_t \mid \mathbf{x}_{t-1}), \quad q(\mathbf{x}_t \mid \mathbf{x}_{t-1}) := \mathcal{N}\left(\mathbf{x}_t; \sqrt{1-\beta_t}\mathbf{x}_{t-1}, \beta_t\mathbf{I}\right) . \quad (3)$$

Let $\alpha_t := 1 - \beta_t$ and $\bar{\alpha}_t := \prod_{s=1}^{t} \alpha_s$. To be noted that the latent variable $\mathbf{x}_t$ can be easily sampled given $\mathbf{x}_0$ and a $\epsilon \sim \mathcal{N}(0, \mathbf{I})$, using the identity $\mathbf{x}_t(\mathbf{x}_0, \epsilon) = \sqrt{\bar{\alpha}_t}\mathbf{x}_0 + \sqrt{1-\bar{\alpha}_t}\epsilon$.

This forward diffusion converges geometrically to a standard Gaussian distribution, so we consider the distribution of the last latent variable as such: $p(\mathbf{x}_{T_{\mathrm{diff}}}) := \mathcal{N}(0, \mathbf{I})$. The reverse process distribution boils down to:

$$p_\theta(\mathbf{x}_{0:T_{\mathrm{diff}}}) := p(\mathbf{x}_{T_{\mathrm{diff}}}) \prod_{t=1}^{T_{\mathrm{diff}}} p_\theta(\mathbf{x}_{t-1} \mid \mathbf{x}_t), \quad p_\theta(\mathbf{x}_{t-1} \mid \mathbf{x}_t) := \mathcal{N}(\mathbf{x}_{t-1}; \boldsymbol{\mu}_\theta(\mathbf{x}_t, t), \beta_t\mathbf{I}) , \quad (4)$$

where $\boldsymbol{\mu}_\theta(\mathbf{x}_t, t) = \frac{1}{\sqrt{\alpha_t}}\left(\mathbf{x}_t - \frac{\beta_t}{\sqrt{1-\bar{\alpha}_t}}\boldsymbol{\epsilon}_\theta(\mathbf{x}_t, t)\right)$, $\boldsymbol{\epsilon}_\theta(\mathbf{x}_t, t)$ being a UNet (Ronneberger et al., 2015) that takes as parameter the latent $\mathbf{x}_t$ and the time-step $t$ and minimizes the following loss:

$$L(\epsilon_\theta) := \sum_{t=1}^{T_{\mathrm{diff}}} \mathbb{E}_{\boldsymbol{x}_0 \sim q(\boldsymbol{x}_0), \epsilon_t \sim \mathcal{N}(\mathbf{0}, \boldsymbol{I})} \left[\left\|\epsilon_\theta\left(\sqrt{\bar{\alpha}_t}\boldsymbol{x}_0 + \sqrt{1-\bar{\alpha}_t}\epsilon_t, t\right) - \epsilon_t\right\|_2^2\right] . \quad (5)$$

Once trained, the diffusion model can generate new data in $\mathbb{R}^D$, starting from $\mathbf{x}_{T_{\mathrm{diff}}} \sim \mathcal{N}(0, \mathbf{I})$ and gradually denoising it using Equation 4. The Denoising Diffusion Implicit Models (DDIM) framework (Song et al., 2020) enables to accelerate this computationally intensive sampling process, keeping only a chosen number of denoising steps and skipping the others.

$D$ being often high (e.g. for images), Rombach et al. (2022) propose to push back the diffusion and the learning of the reverse process into the low-dimensional latent space of a pre-trained VAE, making the training and the sampling faster. They call their method Latent Diffusion Models (LDM).

## 3 Method: Longitudinal Latent Diffusion Models (LLDM)

### 3.1 Framework

Let $(\boldsymbol{x}^i)_{1 \leq i \leq N}$ be our training set of observed *entities* or *individuals* through time, assumed sampled i.i.d. from an unknown distribution $p$ that does not depend on $i$. Each entity $i = 1, \ldots, N$ is a

sequence of (possibly high-dimensional, e.g. images) observations: $\boldsymbol{x}^i = (\boldsymbol{x}_1^i, ...\boldsymbol{x}_{T_i}^i)$, where, for each $j$, $\boldsymbol{x}_j^i \in \mathcal{X} := \mathbb{R}^D$ and $T_i$ being the number of observations for a given individual.

We place ourselves in the VAE framework (see 2.1), adapting it to the longitudinal universe. Let $\boldsymbol{x}^i = (\boldsymbol{x}_1^i, \cdots, \boldsymbol{x}_{T_i}^i)$ be an entity; for each observation $\boldsymbol{x}_j^i$, we denote as $\boldsymbol{z}_j^i$ its embedding, lying in a latent space $\mathcal{Z} := \mathbb{R}^d$, where $d$ is significantly lower than $D$. Weights are shared across all $j$ for the encoder and the decoder.

Finally, we assume that when conditioning on a single $\boldsymbol{z}_j^i$, the observations $(\boldsymbol{x}_j)_j^i$ are not independent. The joint likelihood $p_\theta\left(\boldsymbol{x}_1^i, \cdots, \boldsymbol{x}_{T_i}^i\right) = \int_{\mathcal{Z}} p_\theta\left(\boldsymbol{x}_1^i, \cdots, \boldsymbol{x}_{T_i}^i \mid \boldsymbol{z}_j^i\right) p\left(\boldsymbol{z}_j^i\right) d\boldsymbol{z}_j$ - with $p(\boldsymbol{z}_j^i)$ being a prior on $\boldsymbol{z}_j^i$ - is not factorisable as is. However, when all the latent variables $(\boldsymbol{z}_j^i)_l$ are observed, we assume that the observations are independent and then the joint log-likelihood writes:

$$
\begin{aligned}
\log p_\theta\left(\boldsymbol{x}_1^i, \cdots, \boldsymbol{x}_{T_i}^i\right) &= \log \int_{\mathcal{Z}} \prod_{l=0}^{T_i} p_\theta\left(\boldsymbol{x}_l^i \mid \boldsymbol{z}_l^i\right) p\left(\boldsymbol{z}_j^i\right) d\boldsymbol{z}_j^i \\
&\geq \mathbb{E}_{\boldsymbol{z}_j^i \sim q_\phi(\cdot \mid \boldsymbol{x}_j^i)} \sum_{l=0}^{T_i} \log p_\theta\left(\boldsymbol{x}_l^i \mid \boldsymbol{z}_l^i\right) - D_{\mathrm{KL}}\left(q_\phi\left(\boldsymbol{z}_j^i \mid \boldsymbol{x}_j^i\right) \| p\left(\boldsymbol{z}_j^i\right)\right) := -\mathcal{L}.
\end{aligned}
\tag{6}
$$

Therefore, our goal is to learn once given a $\boldsymbol{z}_j^i$ how to compute the other $\boldsymbol{z}_l^i$: we want to model the dependency structure between the latent variables of each observation of the sequence (and by doing so, model the dependency between the observations themselves).

## 3.2 PRE-TRAINING OF A LDM

As done by Rombach et al. (2022), we first pre-train a vanilla VAE as a first-stage model. Here, we leave the longitudinal universe, and consider all the observations without any sequential dependency. Our training set becomes a set of $\sum_{i=1}^N T_i$ observations. To be noted that, as discussed in 2.1, this VAE yields a Riemannian manifold $\mathcal{M} = (\mathbb{R}^d, \mathbf{G})$, on which we can define a Riemannian uniform distribution: this will be useful in the following section.

To then train the LDM *per se*, we only keep the last observations' embeddings of each entity ($N$ vectors in total) as a training set. We train a diffusion model (see 2.2) in a similar way as in Rombach et al. (2022). The dimension $d$ being reasonably small, the training is not that time consuming.

Once trained, the LDM is able to sample from the last observations' embeddings distribution, by generating trajectories (of length $T_{\mathrm{diff}}$ steps) from a variable sampled from $\mathcal{N}(0, \mathbf{I})$ in $\mathbb{R}^d$.

## 3.3 TRAINING OF THE LVAE

Let us get back to the longitudinal framework, with a training set of $N$ *entities*. We consider a VAE that has the same architecture as the pre-trained VAE in 3.2 - to avoid confusion, we will call it the Longitudinal Variational Autoencoder (LVAE).

To navigate between the latent embeddings and model the dependency structure between them, we use the *forward* and the pre-learned *backward* diffusion processes. In a sense, we have pre-trained trajectories (given by the LDM), and we want now the LVAE to structure the latent embeddings taking these trajectories into account.

We consider a standard Gaussian prior on $\boldsymbol{z}_1^i$, the embedding of the first observation: $p_1(\cdot) := \mathcal{N}(\cdot; 0, \mathbf{I})$. For $2 \leq j \leq T$, we consider the Riemannian uniform prior on $\mathcal{M}$, the manifold yielded by the pre-trained VAE (see 3.2), as a soft prior on the LVAE such that it works as a regularizer in the latent space: $p_j(\cdot) := \mathcal{U}_{\mathrm{Riem}}(\cdot; \mathcal{M})$.

For each position $j = 1, \ldots, T_i$ of the sequence, consider furthermore $t_j^i$ the corresponding diffusion time step in $\{0, \cdots, T_{\mathrm{diff}}\}$ and let $\tau_j^i := t_j^i - t_{j+1}^i$ for $j < T_i$. $(t_j^i)_{1 \leq i \leq N, 1 \leq j \leq T_i}$ represents the matching between the *real timeline* of the longitudinal sequence and the *diffusion timeline*. We set Typically, $t_T^i := 0$ for all $i$. For the individual $i_{\max}$ that has the longest duration between their first and last visit, $t_T^{i_{\max}} = T_{\mathrm{diff}}$. We say that the dataset is *regularly-sampled* if $t_1^i := t_j$ do not depend on $i$. If the observations are temporally equally distributed, $\tau_j := \tau := \lfloor \frac{T_{\mathrm{diff}}}{T-1} \rfloor$ (and then $t_j = (T-j)\tau$).

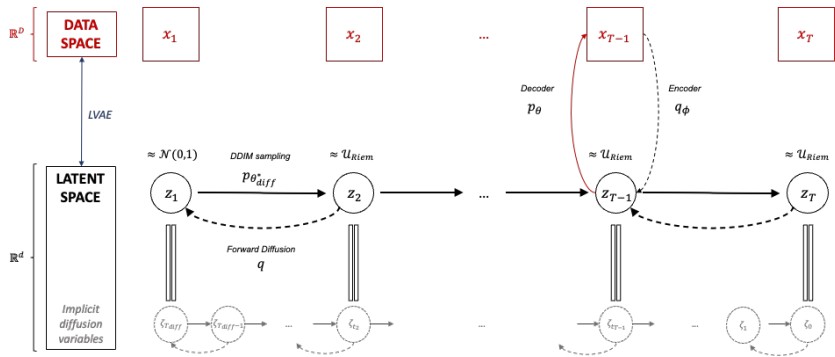

Figure 1: Probabilistic graphical model of the LLDM. Note that the encoder $p_\theta$ and the decoder $q_\phi$ do not depend on the position $j = 1...T$ and that $z_1$ is exactly equal to $\zeta_{T_{\text{diff}}}$, $z_2$ to $\zeta_{t_2}$ and so on.

For a given embedding $\boldsymbol{z}_j^i$, we can sample:

- $\boldsymbol{z}_{j+1}^i$, if $j < T_i$, by making $\tau_j^i$ denoising steps. Thanks to the DDIM (Song et al., 2020) framework this sampling can be done in a single step with the pre-trained diffusion model: this transition distribution, that depends on the matching $(t_j^i)_{1 \leq i \leq N, 1 \leq j \leq T_i}$ (given as an input), will be denoted as $p_{\theta_{\text{diff}}^*}(\boldsymbol{z}_{j+1}^i | \boldsymbol{z}_j^i)$, with a hyperparameter $\eta$ (set to 1 unless stated otherwise) that controls the stochasticity (all details are given in Appendix E).

- $\boldsymbol{z}_{j-1}^i$, if $j > 1$, by making $\tau_{j-1}^i$ forward diffusion steps. This is also done in one step as we know that $q\left(\boldsymbol{z}_{j-1}^i \mid \boldsymbol{z}_j^i\right) = \mathcal{N}\left(\boldsymbol{z}_{j-1}^i; \sqrt{\bar{\alpha}_{t_j^i, t_{j-1}^i}} \boldsymbol{z}_j^i, \left(1 - \bar{\alpha}_{t_j^i, t_{j-1}^i}\right) \mathbf{I}\right)$ with $\beta_1, ..., \beta_{T_{\text{diff}}}$ the variance schedule of the pre-trained LDM, $\alpha_t := 1 - \beta_t$ and $\bar{\alpha}_{t_j^i, t_{j-1}^i} := \prod_{s=t_j^i+1}^{t_{j-1}^i} \alpha_s$.

Figure 1 summarizes this generative model. Algorithm 1 outlines the training procedure for the final component of the LLDM, the LVAE.

---

**Algorithm 1** Training the LVAE

---

**Require:** Training set of sequences $(\boldsymbol{x}_1^i, \ldots, \boldsymbol{x}_{T_i}^i)_{1 \leq i \leq N}$, pre-trained LDM, priors $(p_j)_{j=1...T}$, matching $(t_j^i)_{1 \leq i \leq N, 1 \leq j \leq T_i}$
1: **while** not converged **do**
2:     **for** $i = 1$ to $N$ **do**
3:         Choose $j \in \{1, \ldots, T_i\}$ randomly
4:         $\boldsymbol{z}_j^i \sim q_\phi(\boldsymbol{z}_j^i | \boldsymbol{x}_j^i)$                ▷ Encode (only the observation $j$)
5:         **for** $l = j + 1$ to $T_i$ **do**
6:             Sample $\boldsymbol{z}_l^i \sim p_{\theta_{\text{diff}}^*}(\boldsymbol{z}_l^i | \boldsymbol{z}_{l-1}^i)$     ▷ Propagate into future - Backward Diffusion
7:         **end for**
8:         **for** $l = j - 1$ to $1$ **do**
9:             Sample $\boldsymbol{z}_l^i \sim q(\boldsymbol{z}_l^i | \boldsymbol{z}_{l+1}^i)$         ▷ Propagate into past - Forward Diffusion
10:       **end for**
11:        **for** $l = 1$ to $T_i$ **do**
12:          Sample $\hat{\boldsymbol{x}}_l^i \sim p_\theta\left(\hat{\boldsymbol{x}}_l^i \mid \boldsymbol{z}_l^i\right)$          ▷ Decode the whole sequence
13:        **end for**
14:        $\mathcal{L} = -\frac{1}{T_i} \sum_{l=0}^{T_i} \log p_\theta(\hat{\boldsymbol{x}}_l^i | \boldsymbol{z}_l^i) + \log q_\phi(\boldsymbol{z}_j^i | \boldsymbol{x}_j^i) - \log p_j(\boldsymbol{z}_j^i)$
15:        Take a gradient descent step on $\nabla_\phi \mathcal{L}$ and on $\nabla_\theta \mathcal{L}$
16:     **end for**
17: **end while**

---

### 3.4 SAMPLING

Thanks to this generative model, we are able to propose a simple generation procedure that consists in sampling $\boldsymbol{z}_1 \sim \mathcal{N}(0, \mathbf{I})$ and sequentially sample $\boldsymbol{z}_2 \sim p_{\theta_{\text{diff}}^*}(\boldsymbol{z}_2 | \boldsymbol{z}_1), \ldots, \boldsymbol{z}_T \sim p_{\theta_{\text{diff}}^*}(\boldsymbol{z}_T | \boldsymbol{z}_{T-1})$.

However, as already mentioned in 2.1, the standard normal sampling in the VAE setting is very limited as the observations' embeddings end up being structured as a Riemannian manifold. This led us to consider a more relevant, *geometry aware* generative procedure taking advantage of both these manifold distributions and the stochastic dynamic provided by our diffusion process.

It is detailed in Algorithm 2 for a simplified case with regularly-sampled data and $T_i := T$ for all $i$ - a more complex procedure for irregularly-sampled data and oversampling is detailed in Appendix F. We consider the $T$ manifolds yielded by the LVAE (one for each position). The sampling procedure mimics the training one (Algorithm 1), but the randomly chosen starting position $j$ becomes an input (*start_index*) and instead of encoding a training observation we sample from a Riemannian uniform distribution on the considered manifold $\mathcal{M}_{\text{start\_index}}$.

---

**Algorithm 2** LLDM sampling for regularly-sampled dataset and $T_i := T$ for all $i$

---

**Require:** Trained LLDM, training set $(\boldsymbol{x}^i)_{i=1\dots N}$, length of sequence $T$, start_index $= 1, \dots, T$
1: Compute $\mathbf{G}_{\text{start\_index}}$, the Riemannian metric, using the start_index$^{\text{th}}$ $((\boldsymbol{x}^i_{\text{start\_index}})_{i=1\dots N}$ observations only, let $\mathcal{M}_{\text{start\_index}} = (\mathbb{R}^d, \mathbf{G}_{\text{start\_index}})$ the corresponding manifold
2: Sample $\boldsymbol{z}_{\text{start\_index}} \sim \mathcal{U}_{\text{Riem}}(\mathcal{M}_{\text{start\_index}})$ using a HMC sampler
3: **for** $l = $ start_index $+ 1$ to $T$ **do**
4:     Sample $\boldsymbol{z}_l \sim p_{\theta^*_{\text{diff}}}(\boldsymbol{z}_l|\boldsymbol{z}_{l-1})$                ▷ Propagate into future - Backward Diffusion
5: **end for**
6: **for** $l = $ start_index $- 1$ to $1$ **do**
7:     Sample $\boldsymbol{z}_l \sim q(\boldsymbol{z}_l|\boldsymbol{z}_{l+1})$                      ▷ Propagate into past - Forward Diffusion
8: **end for**
9: **for** $l = 1$ to $T$ **do**
10:     Sample $\hat{\boldsymbol{x}}_l \sim p_\theta(\hat{\boldsymbol{x}}_l \mid \boldsymbol{z}_l)$                     ▷ Decode the whole sequence
11: **end for**
      **return** $(\hat{\boldsymbol{x}}_1, \dots, \hat{\boldsymbol{x}}_T)$

---

## 4 EXPERIMENTS

### 4.1 DATA

We considered three different longitudinal datasets of increasing complexity.

1. *Starmen* is a synthetic longitudinal dataset that consists in 1,000 sequences of 10 (1, 64, 64) images, representing *starmen* raising their left arm and generated according to the diffeomorphic model of Bone et al. (2018). We split the dataset, keeping 800 samples for training, 100 for validation and 100 for test set.

2. The *Sprites* dataset (Reed et al., 2015) consists in sequences of 7 (3, 64, 64) images, representing video games characters performing actions such as walking or dancing. Training set contains 8,000 sequences, validation set 1,000 and test set 2,664.

3. An irregularly-sampled tabular dataset that represents a virtual large-scale cohort, based on the Alzheimer's Disease Neuroimaging Initiative (ADNI) [2]. We extract 4,000 patients with 8 observations, 3,000 that have 7 observations and 3,000 that have 6. We have access to the duration between each visit of a patient. Each observation consists in a vector of 120 features (glucose metabolism (SUVr) projected on the AAL2 atlas). We then randomly split each subset to have in total 8,000 training, 1,000 validation and 1,000 test samples.

For comparison, we tried to use state-of-the-art NODE-based generative method for longitudinal data, such as ODE$^2$VAE (Yildiz et al., 2019) [3], but training fails on these high-dimensional datasets, quickly yielding `NaNs`. Unfortunately, Kanaa et al. (2021) do not provide any code. At the end of the day, we compare our method to the one proposed by Chadebec & Allassonnière (2023) [4], that we call LVAE-NF, and Fortuin et al. (2020), GP-VAE. Each competitor is trained with the same architectures and implementation details - see Appendix A.

---

[2]Available here `https://project.inria.fr/digitalbrain/`
[3]`https://github.com/cagatayyildiz/ODE2VAE`
[4]Code available on request

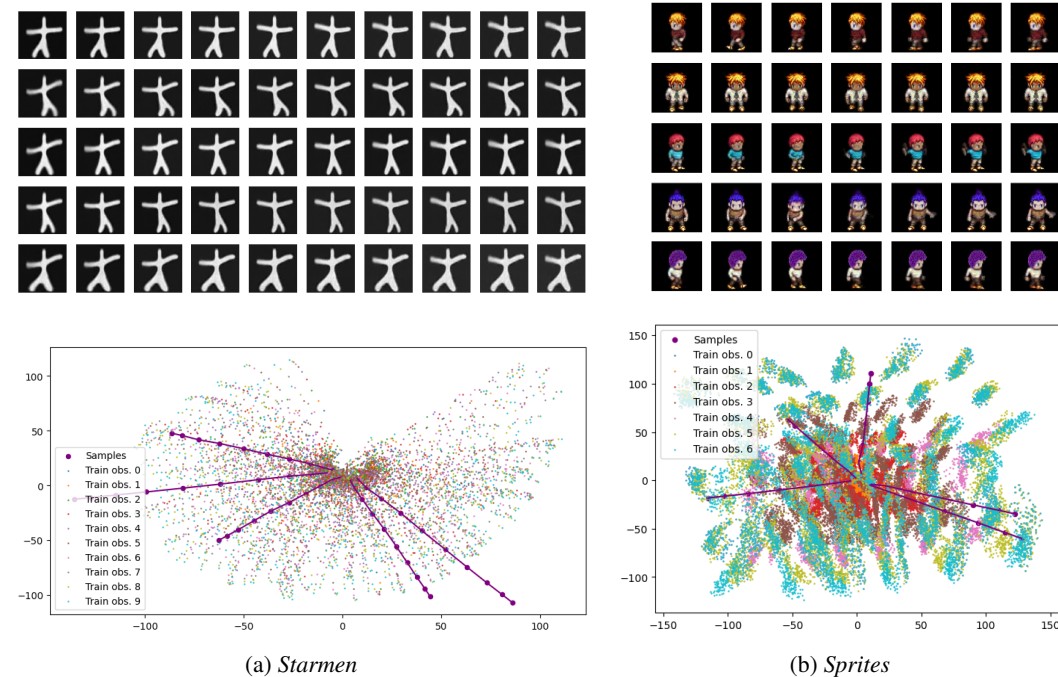

(a) *Starmen*        (b) *Sprites*

Figure 2: Unconditional sequence generation using Algorithm 2 and *start_index = 3* for both datasets. *Top*: generated fully synthetic sequences. *Bottom*: Latent trajectories of the generated sequences. Projection over the two principal components of the trained embeddings. For each $j$, the trained embeddings have been displayed in different colors to show the different manifolds $\mathcal{M}_j$.

| Fréchet Inception Distance (FID) ↓ | Starmen | Sprites |
|---|---|---|
| GP-VAE | 37.5 (0.1) | 60.2 (0.3) |
| LVAE-NF | 42.5 (0.6) | 49.0 (1.2) |
| LLDM | **34.4** (1.7) | **35.8** (0.1) |

Table 1: FID computed on test sets. Averaged over five runs, (·) indicates standard deviation.

## 4.2 GENERATION

**Unconditional generation** We evaluate here the ability of a trained LLDM to generate relevant fully synthetic trajectories. Figure 2 displays examples of five sequences generated, on each dataset.

Moreover, Figures 2a and 2b show the behavior of the LVAE component of the LLDM. For *Sprites*, the latent space is very well organized, displaying clear clusters according to the position in the longitudinal sequence. These clusters are wisely placed according to the diffusion process that is pre-trained. This is due to the fact that this dataset contains clearly different movements (raising arm, walking, dancing) with clearly distinct characters. As for *Starmen*, a less diverse dataset where all the observations within a sequence appear quite similar, the latent space is more monolithic yet still displays a dynamic consistent with the diffusion process. This is a key feature of LLDM, letting access to a latent space that is interpretable and reflects the characteristics of the training dataset.

| Model | LLDM | LVAE-NF | GP-VAE |
|---|---|---|---|
| KL Divergence | **7.97** | 83.37 | 128.27 |

Table 2: KL divergence values between fitted Gaussians on the full (without considering temporal dependence) *ADNI-based* test set and each of the generated sets (same size as test set).

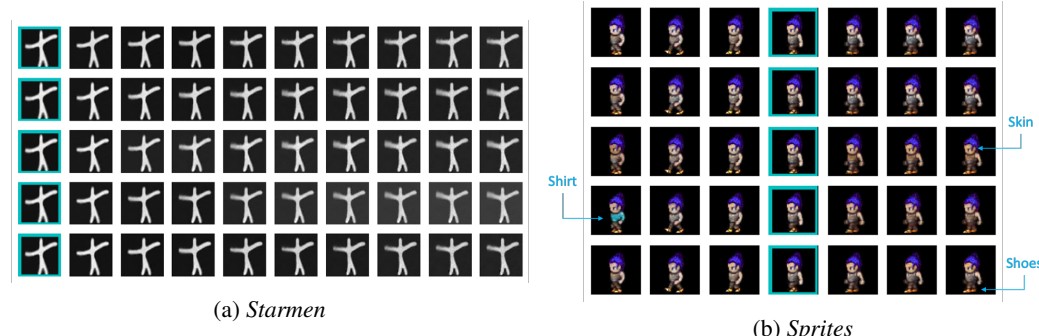

(a) *Starmen*

(b) *Sprites*

Figure 3: Five conditionally generated synthetic sequences. Contoured in cyan is the frozen position.

For *Sprites* and *Starmen*, Table 1 displays the FID metric (computed on all the images without considering temporal dependence), showing that LLDM significantly outperforms its competitors, thanks to diverse (due to its inherent stochasticity) yet faithful generations. For the *ADNI-based* dataset, Table 2 shows that LLDM generated data is the closest to the true one - in a Gaussian analysis. For these two tables, we used Algorithm 2, and chose the *start_index* yielding the best value on validation set - even though this choice has a little impact (Appendix C).

**Conditional generation**    Here, we generate full synthetic sequences again, but we freeze $z_{start\_index}$ (still sampled from a uniform Riemannian distribution on the corresponding manifold), ensuring each sequence shares the same start. A key feature of LLDM is its ability to generate variations in the samples, even under this conditioning. Figure 3 shows examples for *Starmen* and *Sprites*, freezing the first and middle observations, respectively.

In *Starmen* (Figure 3a), variations occur in the final arm position and shape, consistent with training data that shows only one movement. In *Sprites* (Figure 3b), variations are more noticeable, with changes in shirt, skin color, or shoes while maintaining core movement.

These variations are plausible, maintaining the overall movement while allowing for individual differences. This is the kind of variability expected in medical or econometric data, where group trends are preserved but individual variation is allowed.

### 4.3    FUTURE PREDICTION

A notable capability of LLDM is future prediction, starting from the embedding of the last observed image. Controlled by the DDIM sampler's $\eta$ hyperparameter, Figure 4 shows LLDM accurately predicting future sequences over several steps (4a), while also generating diverse outcomes around a core tendency (4b). This contrasts with deterministic methods like LVAE-NF, which yield a single prediction. Appendix G.3 quantifies this variability by computing the variance of MSE over 10 runs, demonstrating that less conditioning leads to more variability in later predictions.

Table 3 computes the MSE between predicted and true observations on the test set. In that challenging setting with less structured and irregularly-sampled data, LLDM is able to achieve more faithful predictions than LVAE-NF, showing its versatility. We provide a similar table for *Sprites* in Appendix G.2 using the Structural Similarity Index Measure (SSIM).

### 4.4    OVERSAMPLING

Linking the real timeline of the sequence to the diffusion timeline enables us to discretize even more the duration between each observation: LLDM is able to increase the frequency of a sequence by successfully imputing the intermediary steps. This oversampling can be done by decoding unseen latent variables from the diffusion models, $\zeta_k$ for $k \in \bigcup_{j=1}^{T-1}]t_j, t_{j+1}[$ (Figure 1). Figure 5a shows an example when decoding the $\zeta_{\frac{t_j+t_{j+1}}{2}}$, doubling the frequency of a given sequence. Other generative competitors can not achieve that, to the best of our knowledge.

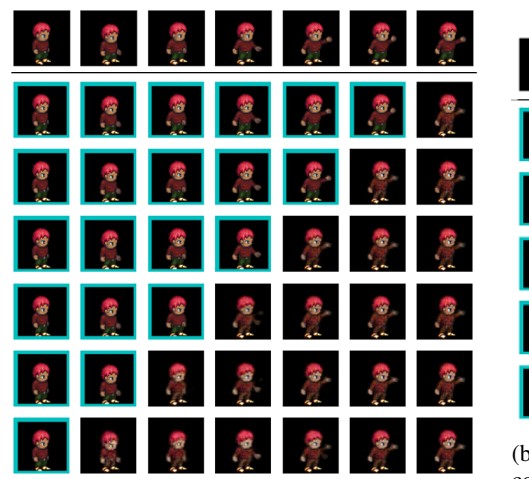

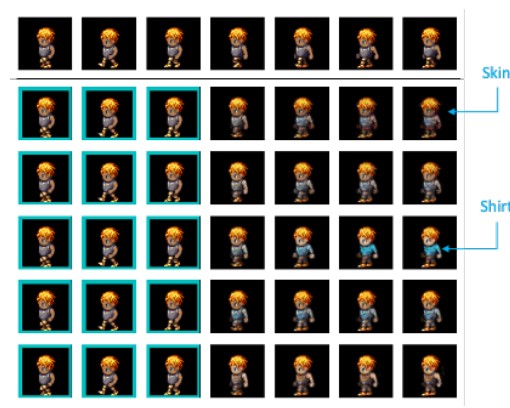

(b) Fixed number of prediction steps. Variations around core tendency. DDIM $\eta$ is increased to 5.

(a) Varying number of prediction steps.

Figure 4: Future prediction with LLDM. At the top is the true sequence, contoured in cyan are the images that are given (not predicted).

| Number of predicted steps | | Obs 1 | Obs 2 | Obs 3 | Obs 4 | Obs 5 | Obs 6 | Obs 7 | Obs 8 |
|---|---|---|---|---|---|---|---|---|---|
| 1 | GP-VAE | - | - | - | - | - | - | - | 16.82 |
| | LVAE-NF | - | - | - | - | - | - | - | 24.52 |
| | LLDM | - | - | - | - | - | - | - | **6.89** |
| 2 | GP-VAE | - | - | - | - | - | - | 16.61 | 17.08 |
| | LVAE-NF | - | - | - | - | - | - | 22.57 | 56.07 |
| | LLDM | - | - | - | - | - | - | **5.07** | **10.73** |
| 3 | GP-VAE | - | - | - | - | - | 16.71 | 16.64 | 16.73 |
| | LVAE-NF | - | - | - | - | - | 14.77 | 28.23 | 63.14 |
| | LLDM | - | - | - | - | - | **4.75** | **7.70** | **13.39** |
| 4 | GP-VAE | - | - | - | - | 16.78 | 16.61 | 16.97 | 16.71 |
| | LVAE-NF | - | - | - | - | 14.42 | 14.38 | 28.46 | 66.45 |
| | LLDM | - | - | - | - | **4.76** | **6.87** | **9.53** | **14.07** |
| 5 | GP-VAE | - | - | - | 17.36 | 17.04 | 16.70 | 16.61 | 16.88 |
| | LVAE-NF | - | - | - | 15.22 | 14.60 | 14.69 | 29.42 | 69.07 |
| | LLDM | - | - | - | **4.80** | **6.24** | **7.74** | **9.75** | **13.39** |
| 6 | GP-VAE | - | - | 17.00 | 17.56 | 16.77 | 16.70 | 16.64 | 16.88 |
| | LVAE-NF | - | - | 15.24 | 15.26 | 14.67 | 14.89 | 30.32 | 68.93 |
| | LLDM | - | - | **4.98** | **6.01** | **6.93** | **8.00** | **9.50** | **12.16** |
| 7 | GP-VAE | - | 17.16 | 17.00 | 17.27 | 16.80 | 16.70 | 16.62 | 16.70 |
| | LVAE-NF | - | 14.28 | 14.33 | 14.28 | 13.98 | 14.14 | 28.07 | 66.08 |
| | LLDM | - | **5.12** | **5.74** | **6.32** | **6.87** | **7.54** | **8.52** | **10.16** |

Table 3: Mean squared error on test set between predicted and true steps for *ADNI-based* dataset. Average over five runs. Standard deviations are given in Appendix G.3. Starting from the last seen embedding for LLDM and LVAE-NF, masking with zeroes the unseen data for GP-VAE.

It is also possible to easily generate fully synthetic oversampled sequences (with more granular time steps) by adapting algorithm 2 - see Figure 5b. Details are provided in Appendix F.

## 4.5 ROBUSTNESS TO MISSING TRAINING DATA

In the context of longitudinal data, we often encounter poor data quality, particularly in the form of *missing observations*. Figure 6 demonstrates that LLDM (like LVAE-NF) is able to maintain its performance, even when up to 40 % of training and validation observations are removed. In contrast, GP-VAE is significantly less robust, with a striking drop in performance.

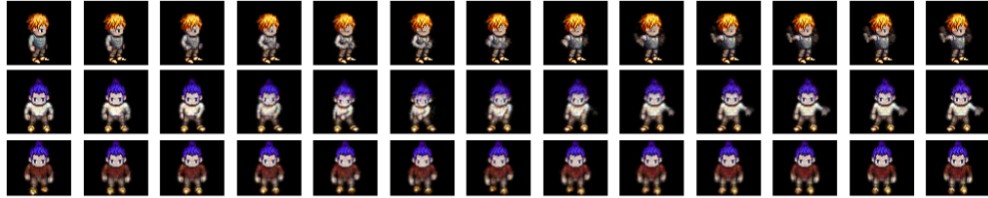

(a) *Top*: original. *Bottom:* Oversampled sequence, doubled frequency. Contoured in red are the initial time steps (that are reconstructed), others are imputed. The arm swinging movement *and* the walking appear more continuous, more fluid.

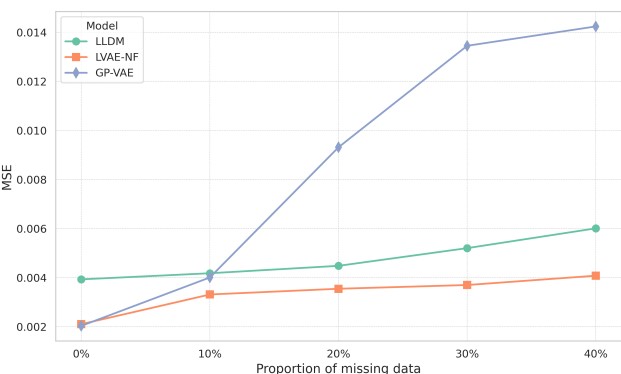

(b) Oversampled (doubled frequency) fully synthetic generated sequences. *Top:* Eye closing and arm movements (especially elbow folding). *Middle:* Arm and leg movements. *Bottom:* Walking.

Figure 5: Oversampling with LLDM.

Figure 6: Mean squared error (averaged over all pixels) on the *Sprites* test set when varying proportions of training and validation data are randomly removed, with models re-trained for each case. For LLDM and LVAE-NF, sampling of $j$ is restricted to available indexes. For GP-VAE, a zero mask is used to hide unavailable data.

## 5 CONCLUSION

We present the Longitudinal Latent Diffusion Model (LLDM), a generative approach for high-dimensional longitudinal data that combines VAE embeddings with a latent diffusion process, offering remarkable flexibility. By decoding diffusion latent variables at specific time steps, LLDM leverages latent space trajectories rather than merely blurring images like traditional diffusion models. LLDM generates diverse, realistic longitudinal trajectories both unconditionally and conditionally. The alignment of the real timeline with the granular diffusion timeline enables tasks such as future prediction, oversampling, and imputation, with controlled stochasticity as a key feature. Its efficiency allows for successful training on limited or incomplete datasets.

Future research could explore theoretical guarantees that demonstrate the convergence between the modeled and true distributions. Additionally, the use of Riemannian diffusion models (Bortoli et al., 2022; Huang et al., 2022) within the VAE latent space appears promising, as these models may better align with the Riemannian geometry of the space, potentially improving generation quality. However, the applicability of these methods in (relatively) high-dimensional settings remain to be explored, as they have primarily been applied in 2D or 3D manifolds (e.g., torus, hypersphere).

ACKNOWLEDGMENTS

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

# APPENDIX

## A  ARCHITECTURE AND IMPLEMENTATION DETAILS

**Architectures**  Table 4 summarizes the used architectures. These are mainly convolutional for images and MLP-based for tabular data.

| Dataset | Starmen | Sprites | ADNI |
|---|---|---|---|
| Input Dimension | (1, 64, 64) | (3, 64, 64) | (1, 120) |
| | Encoder | | |
| | Conv2D(1, 16, 4, 2) | Conv2D(3, 16, 4, 2) | Linear(120, 60) |
| | Conv2D(16, 32, 4, 2) | Conv2D(16, 32, 4, 2) | Linear(60, 30) |
| | LeakyReLU | LeakyReLU | ReLU |
| | Conv2D(32, 64, 3, 2) | Conv2D(32, 64, 3, 2) | Linear(30, 15) |
| | LeakyReLU | LeakyReLU | ReLU |
| | LeakyReLU | LeakyReLU | ReLU |
| | 6 ResBlocks | 6 ResBlocks | Linear(15, 9) |
| | Linear(2048, 2x12) | Linear(2048, 2x12) | Linear(9, 2x9) |
| Input Dimension | 12 | 12 | 9 |
| | Decoder | | |
| | Linear(2048) | Linear(2048) | Linear(9, 15) |
| | ConvT(128, 3, 2) | ConvT(128, 3, 2) | ReLU |
| | 6 ResBlocks | 6 ResBlocks | Linear(15, 30) |
| | ConvT(64, 5, 2) | ConvT(64, 5, 2) | ReLU |
| | LeakyReLU | LeakyReLU | Linear(30, 60) |
| | ConvT(32, 5, 2) | ConvT(32, 5, 2) | ReLU |
| | LeakyReLU | LeakyReLU | Linear(60, 120) |
| | ConvT(16, 4, 2) | ConvT(16, 4, 2) | ReLU |
| | LeakyReLU | LeakyReLU | Linear(120, 1) |
| | ConvT(1, 4, 2) | ConvT(3, 4, 2) | - |
| Num. of parameters | $1.07 \cdot 10^6$ | $1.08 \cdot 10^6$ | 19,653 |

Table 4: LVAE architectures for *Starmen*, *Sprites*, and *ADNI-based* datasets. Note that we use the same architectures for the first-stage model of the pre-trained LDM. No normalizing flows were used to enhance the variational posterior, and the prior was a classic standard Gaussian.

| Input Dimension | (1, 3, 2, 2) | (1, 1, 3, 3) |
|---|---|---|
| | Linear(1, 256), SiLU | Linear(1, 128), SiLU |
| | Linear(256, 256) | Linear(128, 128) |
| | Conv2d(1, 64, 3, 1) | Conv2d(1, 32, 3, 1) |
| | **4x** ResBlock(64, 64) | **4x** ResBlock(32, 32) |
| | SpatialTransformer(64) | SpatialTransformer(32) |
| | GroupNorm32, SiLU | GroupNorm32, SiLU |
| | Conv2d(64, 3, 3, 1) | Conv2d(32, 1, 3, 1) |
| Output Dimension | (1, 3, 2, 2) | (1, 1, 3, 3) |
| Num. of parameters | $2.22 \cdot 10^6$ | 353,953 |

Table 5: Denoising UNet architecture for the pre-trained LDM. *Left*: *Starmen* and *Sprites*. *Right*: *ADNI-based* dataset.

**Code**  The code reproducing our results is available at the following (anonymized, for now) link
https://anonymous.4open.science/r/LLDM-C92C. It is based on the PyTorch framework (Ansel et al., 2024). For VAE/LVAE architectures and training, we used the Pythae library

(Chadebec et al., 2022). For diffusion models architecture and DDIM sampler, we used the implementation from `nn.labml.ai` ; and the PyTorch Lightning (Falcon & The PyTorch Lightning team, 2019) framework for their training.

**Training**   Models were trained on a NVIDIA RTX A2000 12GB GPU. We used an Adam optimizer, with learning rate $1e - 3$, a `ReduceLROnPlateau` scheduler (factor 0.5 and patience 4 epochs). Models were trained for 200 epochs with a batch size of 64 sequences (for irregularly-sampled datasets, we batch on subset that are regularly-sampled). The model yielding the best evaluation loss was kept.

For the record, the training times on *Sprites* were, approximately:

- For LLDM: 20mn for the first-stage VAE, 10 mn for the diffusion *per se* and 1h for the LVAE training, so **1h30** in total.
- LVAE-NF: **1h15**
- GP-VAE : **1h20**

## B   SAMPLING TIME

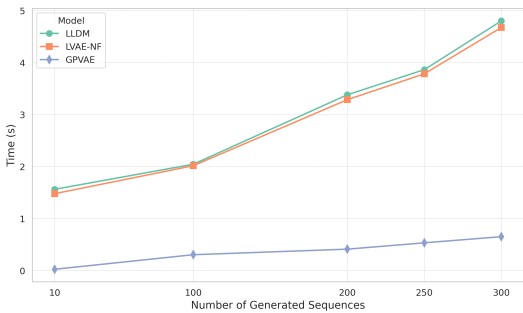

Figure 7: Generation time on *Sprites*: comparison between LLDM and LVAE-NF.

Figure 7 shows that LLDM is able to be on par with competitors in terms of sampling time, being light enough to generate 300 RGB images of size $(64, 64)$ in less than 5 seconds. GP-VAE is way quicker but produces samples of lower quality and without temporal coherence within a sequence.

## C   *start_index* IMPACT ON UNCONDITIONAL GENERATION

| *start_index* | 1 | 2 | 3 | 4 | 5 | 6 | 7 |
|---|---|---|---|---|---|---|---|
| FID | 43.1 | 37.2 | 35.9 | 35.7 | 36.7 | 37.2 | 37.7 |

Table 6: Unconditional generation metrics on *Sprites*, on a single run, varying *start_index*.

| *start_index* | 1 | 2 | 3 | 4 | 5 | 6 | 7 | 8 | 9 | 10 |
|---|---|---|---|---|---|---|---|---|---|---|
| FID | 50.2 | 38.4 | 36.6 | 35.3 | 39.1 | 34.2 | 33.9 | 33.6 | 35.3 | 34.5 |

Table 7: Unconditional generation metrics on *Sprites*, on a single run, varying *start_index*.

We observe that, all in all, LLDM generation remains quite robust to the choice of *start_index*, with slight better performance when choosing to start in the middle of the sequence.

## D   HMC SAMPLER FOR RIEMANNIAN UNIFORM DISTRIBUTION

Given any manifold $\mathcal{M} = (\mathbb{R}^d, \mathbf{G})$, we use a Hamiltonian Monte-Carlo sampler to sample from the Riemannian distribution, which density $p_{\text{target}}(\cdot) := \mathcal{U}_{\text{Riem}}(\cdot; \mathcal{M})$ is given by Equation 2.

Let us define the following Hamiltonian (Duane et al., 1987; Leimkuhler & Reich, 2005):

$$H(z, v) = -\log p_{\text{target}}(z) + \frac{1}{2} v^\top v, \tag{7}$$

where $z \in \mathcal{M}$ is seen as the *position* of a particle traveling on $\mathcal{M}$ and $v \sim \mathcal{N}(0, \mathbf{G}(z))$ as its *velocity*. The Hamiltonian represents then the sum of its potential and kinetic energy.

The evolution in time of such a particle is governed by Hamilton's equations:

$$\begin{cases} \frac{\partial H(z,v)}{\partial v} = v \\ \frac{\partial H(z,v)}{\partial z} = -\nabla_z \log p_{\text{target}}(z) \end{cases} \tag{8}$$

Mimicking the behavior of this particle, the HMC sampler creates a Markov chain of length $n$ $(z_i)_{i=1\dots n}$. Starting from $z_0$, an initial *velocity* is sampled $v_0 \sim \mathcal{N}(0, \mathbf{G}(z_0))$. Then, a proposal $(\tilde{z}, \tilde{v})$ is computed by running $K$ times the following discretization scheme known as the *leapfrog* integrator:

$$\begin{cases} v\left(t + \frac{\varepsilon_{\text{lf}}}{2}\right) = v(t) + \frac{\varepsilon_{\text{lf}}}{2} \cdot \nabla_z \log p_{\text{target}}\left(z(t)\right), \\ z\left(t + \varepsilon_{\text{lf}}\right) = z(t) + \varepsilon_{\text{lf}} \cdot v\left(t + \frac{\varepsilon_{\text{lf}}}{2}\right), \\ v\left(t + \varepsilon_{\text{lf}}\right) = v\left(t + \frac{\varepsilon_{\text{lf}}}{2}\right) + \frac{\varepsilon_{\text{lf}}}{2} \cdot \nabla_z \log p_{\text{target}}\left(z\left(t + \varepsilon_{\text{lf}}\right)\right), \end{cases} \tag{9}$$

where $\varepsilon_{\text{lf}}$ is the leapfrog step size. The proposal is then accepted with probability $\alpha = \min(1, \exp(H(z, v) - H(\tilde{z}, \tilde{v})))$, otherwise $z_1$ stays in $z_0$. We iterate so forth until having $z_n$. It was shown that the chain converges to its stationary distribution $p_{\text{target}}$ (Duane et al., 1987; Liu, 2009; Neal, 2012).

## E   CONSIDERATIONS ON DDIM

We operate here a slight change of notations compared to section 2.2. Let $\gamma_t := \prod_{t=1}^{T_{\text{diff}}} \alpha_t$. Therefore, in Figure 1, the *forward process* (Equation 3) between the diffusion latent variables $(\boldsymbol{\zeta}_t)_{t=1\dots T_{\text{diff}}}$ becomes:

$$q\left(\boldsymbol{\zeta}_t \mid \boldsymbol{\zeta}_{t-1}\right) := \mathcal{N}\left(\boldsymbol{\zeta}_t; \sqrt{\frac{\gamma_t}{\gamma_{t-1}}} \boldsymbol{\zeta}_{t-1}, \left(1 - \frac{\gamma_t}{\gamma_{t-1}}\right) \boldsymbol{I}\right).$$

Noting that the matching $(t_j^i)_{\substack{1 \le i \le N \\ 1 \le j \le T_i}}$ is provided, and that $\boldsymbol{\zeta}_{t_j^i} \equiv \mathbf{z}_j^i$ (with $t_j^i$ decreasing with $j$), we remind that we have the following property that enables us to make "jumps":

$$q\left(\mathbf{z}_{j-1}^i \mid \mathbf{z}_j^i\right) = \mathcal{N}\left(\mathbf{z}_{j-1}^i; \sqrt{\frac{\gamma_{t_{j-1}^i}}{\gamma_{t_j^i}}} \mathbf{z}_j^i, \left(1 - \frac{\gamma_{t_{j-1}^i}}{\gamma_{t_j^i}}\right) \mathbf{I}\right).$$

For the *backward process*, once the diffusion model trained, the DDPM framework (Ho et al., 2020) makes $T_{\text{diff}}$ the following transitions (adapted from Equation 4):

$$p_{\theta_{\text{diff}}^*}(\boldsymbol{\zeta_{t-1}}|\boldsymbol{\zeta_t}) = \mathcal{N}\left(\boldsymbol{\zeta}_{t-1}; \boldsymbol{\mu}_{\theta_{\text{diff}}^*}\left(\boldsymbol{\zeta}_t, t\right), (1 - \frac{\gamma_t}{\gamma_{t-1}})\mathbf{I}\right).$$

To sample $\boldsymbol{z}_{j+1}^i \equiv \boldsymbol{\zeta}_{t_{j+1}^i}$ from a given $\boldsymbol{z}_j^i \equiv \boldsymbol{\zeta}_{t_j^i}$ is time-consuming as it requires $\tau_j^i := t_j^i - t_{j+1}^i$ denoising steps, and the same number of neural function evaluations of the denoising UNet.

The DDIM framework (Song et al., 2020) simplifies this process by enabling to skip transitions (and make "jumps", as in the forward process). For $j = 1...T - 1$, it gives an immediate transition distribution from $\zeta_{t_j^i}$ to $\zeta_{t_{j+1}^i}$ (recall that $t_{j+1}^i < t_j^i$):

$$p_{\theta_{\text{diff}}^*}(\boldsymbol{z_{j+1}^i}|\boldsymbol{z_j^i}) \equiv p_{\theta_{\text{diff}}^*}(\boldsymbol{\zeta_{t_{j+1}^i}}|\boldsymbol{\zeta_{t_j^i}})$$

$$:= \mathcal{N}\left(\mathbf{z}_{\boldsymbol{t_{j+1}^i}};\; \sqrt{\gamma_{t_{j+1}^i}}\left(\frac{\mathbf{z}_j^i - \sqrt{1 - \gamma_{t_j^i}}\,\epsilon_{\theta_{\text{diff}}^*}\left(\mathbf{z}_j^i\right)}{\sqrt{\gamma_{t_j^i}}}\right) + \sqrt{1 - \gamma_{t_{j+1}^i} - \sigma_{t_j^i}^2}\cdot\epsilon_{\theta_{\text{diff}}^*}\left(\mathbf{z}_j^i\right),\, \sigma_{t_j^i}^2(\eta)\mathbf{I}\right),$$

where $\sigma_{t_j^i}(\eta) = \eta\sqrt{\left(1 - \gamma_{t_{j+1}^i}\right)/\left(1 - \gamma_{t_j^i}\right)}\sqrt{1 - \gamma_{t_j^i}/\gamma_{t_{j+1}^i}}$. The hyperparameter $\eta \geq 0$ controls the stochasticity of the sampling by increasing/decreasing the variance ; especially, $\eta = 0$ makes the process completely deterministic.

## F    DETAILS ON OVERSAMPLED GENERATION

For oversampled generation, we just have to change the DDIM sampler's time steps. Instead of only sampling at $(t_j)_j$, we can use custom $K$ strictly decreasing time steps $(\hat{t}_i)_{i=1...K} \in \{0, \ldots, T_{\text{diff}}\}^K$. The sole caveat is that, now, start_index is in $\{1, \ldots, K\}$: therefore, if $\hat{t}_{\text{start\_index}}$ does not correspond to a previous $(t_j)_j$ on which the LLDM has been trained, the manifold $\mathcal{M}_{\text{start\_index}}$ *does not exist*. Therefore, we simply need to enforce that $\hat{t}_{\text{start\_index}}$ is exactly equal to a $t_{j_{\text{start}}}$ with $j_{\text{start}}$ in $\{1, \ldots, T\}$. An updated algorithm is given thereafter (Algorithm 3). This algorithm enables also to adapt Algorithm 2 for irregularly-sampled dataset, the caveat remaining.

---

**Algorithm 3** LLDM oversampled sampling

---

**Require:** Trained LLDM, training set $(x^i)_{i=1...N}$, custom time steps $(\hat{t}_i)_{i=1...K}$, start_index = 1...K
1: Enforce that $\exists j_{\text{start}} = 1...T, \hat{t}_{\text{start\_index}} \equiv t_{j_{\text{start}}}$
2: Compute $\mathbf{G}_{j_{\text{start}}}$, the Riemannian metric, using the $j_{\text{start}}^{\text{th}}$ observations $(\boldsymbol{x}_{j_{\text{start}}}^i)_{i=1...N}$ only, let
$\quad\ \mathcal{M}_{j_{\text{start}}} = (\mathbb{R}^d, \mathbf{G}_{j_{\text{start}}})$ the corresponding manifold
3: Sample $\boldsymbol{z}_{j_{\text{start}}} \sim \mathcal{U}_{\text{Riem}}\left(\mathcal{M}_{j_{\text{start}}}\right)$ using a HMC sampler
4: **for** $l = \text{start\_index} + 1$ to $K$ **do**
5: $\quad$ Sample $\boldsymbol{z}_l \sim p_{\theta_{\text{diff}}^*}(\zeta_{\hat{t}_l}|\zeta_{\hat{t}_{l-1}})$ $\qquad\qquad\qquad$ ▷ Propagate into future - Backward Diffusion
6: **end for**
7: **for** $l = \text{start\_index} - 1$ to 1 **do**
8: $\quad$ Sample $\boldsymbol{z}_l \sim q(\zeta_{\hat{t}_l}|\zeta_{\hat{t}_{l+1}})$ $\qquad\qquad\qquad\qquad$ ▷ Propagate into past - Forward Diffusion
9: **end for**
10: **for** $l = 1$ to $K$ **do**
11: $\quad$ Sample $\hat{\boldsymbol{x}}_l \sim p_\theta\left(\hat{\boldsymbol{x}}_l \mid \boldsymbol{z}_l\right)$ $\qquad\qquad\qquad\qquad\qquad\quad$ ▷ Decode the whole sequence
12: **end for**
$\qquad$ **return** $(\hat{\boldsymbol{x}}_1, \ldots, \hat{\boldsymbol{x}}_T)$

---

## G    ADDITIONAL EXPERIMENTS

### G.1    UNCONDITIONAL GENERATION ON *ADNI-based* DATASET

Figure 8 shows the histogram of a randomly selected number of coordinates (out of 120 total) on real and generated samples. It shows that LLDM is able to catch the mode and shape of the distribution, while for LVAE-NF, these distributions appear left-skewed. On the other hand, GP-VAE captures the mode, but fails to yield a diverse distribution.

Figure 9 shows the final LVAE latent space, which appears less structured than in Figures 2b and 2a due to low variation both within and between sequences. The latent space, once again, reveals insights about the dataset: instead of expanding beyond the $\mathcal{N}(0, \mathbf{I})$ ellipsoid, the final observations remain tightly clustered within it, with diffusion trajectories moving inward rather than outward, as

918
919
920

seen in previous experiments. Despite this challenge, LLDM still achieves strong generation and prediction performance.

921
922
923
924
925
926
927
928
929
930
931
932
933
934
935
936
937
938
939
940
941
942
943
944
945
946
947
948
949
950

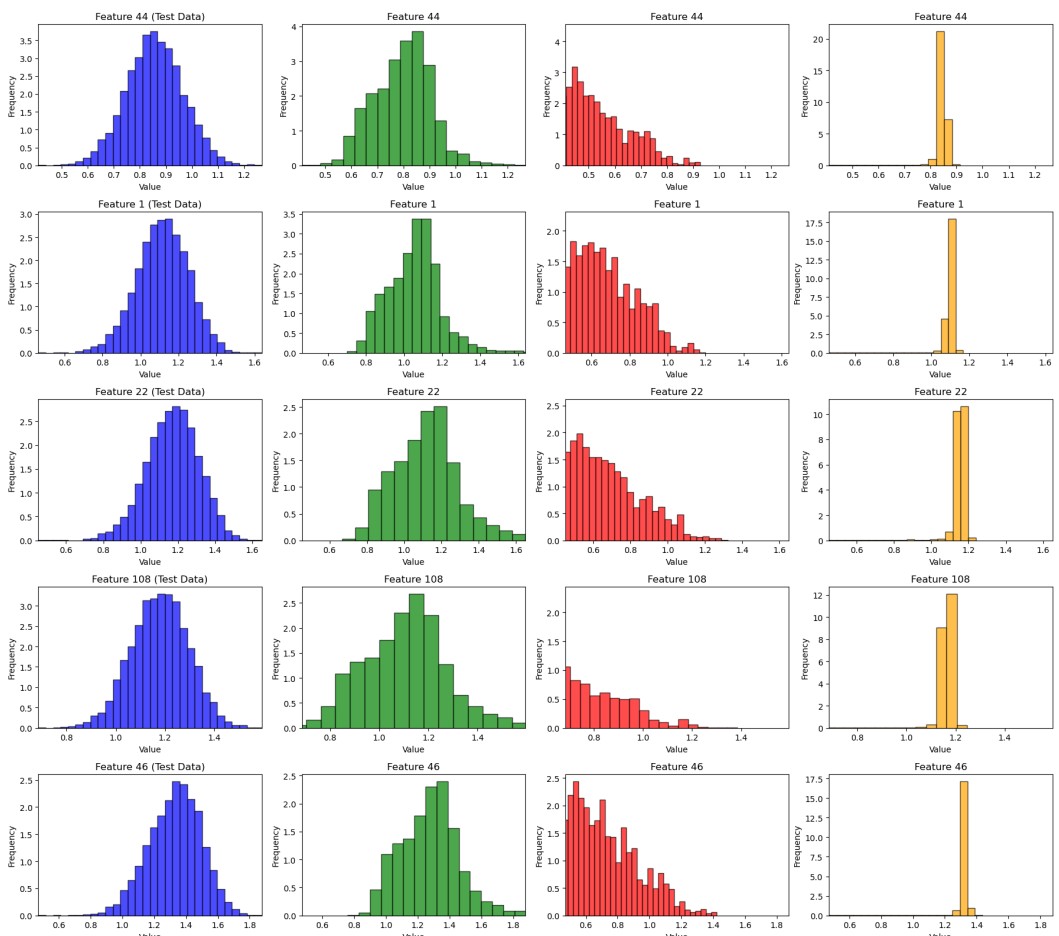

951
952
953
954

Figure 8: Histograms of five randomly sampled features (out of 120 total), comparing the true test data and generated sequences of the same size. The histograms illustrate the distribution of values across the different datasets. *Blue*: Test dataset, *Green*: LLDM, *Red*: LVAE-NF, *Orange*: GP-VAE.

955
956
957
958
959
960
961
962
963
964
965
966
967
968
969
970
971

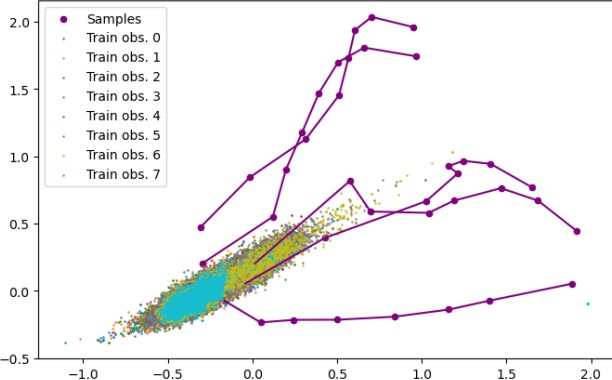

Figure 9: Latent trajectories of five generated sequences. Projection over the two principal components of the trained embeddings. For each $j$, the trained embeddings have been displayed in different colors to show the different manifolds $\mathcal{M}_j$.

## G.2 FUTURE PREDICTION FOR *Sprites*

| Number of predicted steps | | Obs 1 | Obs 2 | Obs 3 | Obs 4 | Obs 5 | Obs 6 | Obs 7 |
|---|---|---|---|---|---|---|---|---|
| 1 | LVAE-NF | - | - | - | - | - | - | 0.94 |
| | LLDM | - | - | - | - | - | - | 0.89 |
| | *GP-VAE* | - | - | - | - | - | - | *0.68* |
| 2 | LVAE-NF | - | - | - | - | - | 0.94 | 0.93 |
| | LLDM | - | - | - | - | - | 0.88 | 0.87 |
| 3 | LVAE-NF | - | - | - | - | 0.94 | 0.93 | 0.94 |
| | LLDM | - | - | - | - | 0.89 | 0.87 | 0.90 |
| 4 | LVAE-NF | - | - | - | 0.94 | 0.93 | 0.94 | 0.93 |
| | LLDM | - | - | - | 0.89 | 0.87 | 0.90 | 0.89 |
| 5 | LVAE-NF | - | - | 0.94 | 0.93 | 0.94 | 0.93 | 0.93 |
| | LLDM | - | - | 0.89 | 0.87 | 0.90 | 0.89 | 0.91 |
| 6 | LVAE-NF | - | 0.89 | 0.89 | 0.89 | 0.89 | 0.91 | 0.91 |
| | LLDM | - | 0.85 | 0.83 | 0.86 | 0.87 | 0.89 | 0.90 |

Table 8: SSIM score on test set between predicted and true steps for *Sprites*. Average over five runs. Standard deviation is negligible.

In Table 8, LLDM is not the best performer but is on par with LVAE-NF *while* adding a key feature: variations around a core tendency (see next section G.3). We note that the GP-VAE do not react well to the zero-masking and yield very low quality generated samples: we only provide the SSIM metric for a first-step prediction.

## G.3 VARIABILITY OF FUTURE PREDICTION AROUND A CORE TENDENCY

| Number of predicted steps | Obs 1 | Obs 2 | Obs 3 | Obs 4 | Obs 5 | Obs 6 | Obs 7 |
|---|---|---|---|---|---|---|---|
| 1 | - | - | - | - | - | - | 0.01 |
| 2 | - | - | - | - | - | 0.17 | 0.13 |
| 3 | - | - | - | - | 0.34 | 0.31 | 0.27 |
| 4 | - | - | - | 0.57 | 0.72 | 0.58 | 0.50 |
| 5 | - | - | 0.76 | 1.47 | 1.49 | 1.27 | 1.08 |
| 6 | - | 6.74 | 8.77 | 10.55 | 13.88 | 13.60 | 14.23 |

Table 9: Standard deviation over ten runs when computing MSE on test set with LLDM on *Sprites*.

| Number of predicted steps | | Obs 1 | Obs 2 | Obs 3 | Obs 4 | Obs 5 | Obs 6 | Obs 7 | Obs 8 |
|---|---|---|---|---|---|---|---|---|---|
| 1 | GP-VAE | - | - | - | - | - | - | - | 0.10 |
| | LLDM | - | - | - | - | - | - | - | 0.01 |
| 2 | GP-VAE | - | - | - | - | - | - | 0.08 | 0.10 |
| | LLDM | - | - | - | - | - | - | 0.04 | 0.04 |
| 3 | GP-VAE | - | - | - | - | - | 0.11 | 0.10 | 0.17 |
| | LLDM | - | - | - | - | - | 0.06 | 0.12 | 0.12 |
| 4 | GP-VAE | - | - | - | - | 0.12 | 0.13 | 0.15 | 0.08 |
| | LLDM | - | - | - | - | 0.06 | 0.16 | 0.14 | 0.12 |
| 5 | GP-VAE | - | - | - | 0.22 | 0.12 | 0.32 | 0.18 | 0.04 |
| | LLDM | - | - | - | 0.06 | 0.13 | 0.20 | 0.18 | 0.19 |
| 6 | GP-VAE | - | - | 0.10 | 0.10 | 0.08 | 0.10 | 0.06 | 0.14 |
| | LLDM | - | - | 0.03 | 0.06 | 0.26 | 0.38 | 0.41 | 0.34 |
| 7 | GP-VAE | - | 0.12 | 0.13 | 0.10 | 0.15 | 0.10 | 0.12 | 0.12 |
| | LLDM | - | 0.17 | 0.16 | 0.19 | 0.22 | 0.32 | 0.36 | 0.43 |

Table 10: Standard deviations over five runs when computing MSE on test set with LLDM and GP-VAE on *ADNI-based* dataset. LVAE-NF has negligible standard deviations. See Table 3 for MSE average values.

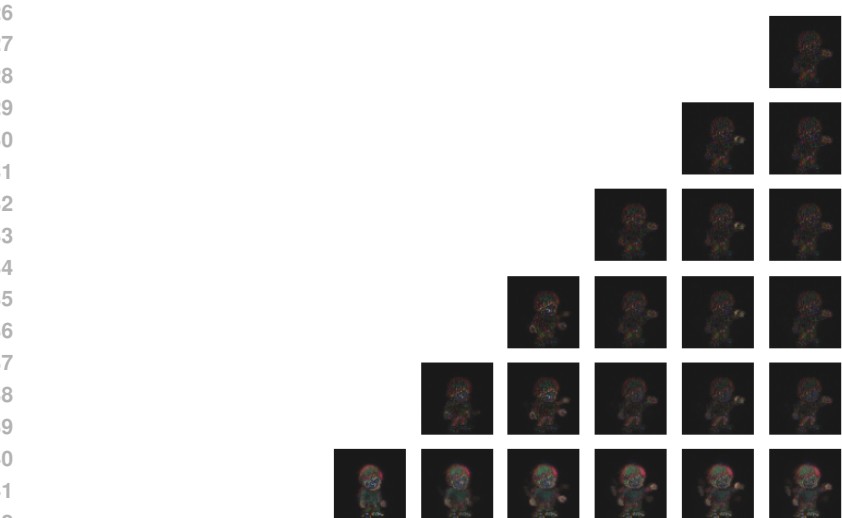

Figure 10: Pixel-wise absolute error between predicted and true observations. Replica of Figure 4a (with the same character). DDIM $\eta$ increased to 5. Average over 10 independent predictions.

In both use cases, as expected, the earlier you condition, the more diverse are the final states. Figure 10 shows that these variations are localized around the character, especially the hand and pants.

