# OpenReview forum: "Longitudinal Latent Diffusion Models"
_ICLR.cc/2025/Conference — Submitted to ICLR 2025_

### Official Review · Reviewer_MJv9 · 2024-11-02

**Soundness:** 3
**Presentation:** 3
**Contribution:** 3
**Rating:** 5
**Confidence:** 3

**Summary:**

The authors add longitudinal component to traditional diffusion models. This requires modeling sequences of images, but unlike in say video, the observations can be far apart in time. The sequence dimension in general is expected to be sparse. The authors tailor a specific pipeline to the setting: the "inner loop" (image generation essentially) is a standard VAE, and the "outer loop" - sequence dimension - trains diffusion in the latent space of the first VAE. The two are combined using so-called Longitudinal VAE (LVAE), which uses the same architecture as the first stage VAE and uses both forward and backward diffusion.

**Strengths:**

I find the setting compelling and useful for real world applications. I like how every design choice is well motivated. Anecdotal results in Figure 2 look persuasive, which makes me believe that the proposed method does work.

**Weaknesses:**

My main concern is about the benchmarks. The authors admit that they couldn't make some of the relevant models work. The ones for which the numbers are reported still appear undertuned: in Table 3, for instance, the proposed model exhibits a strong - and sensible! - pattern, while the other 2 models show no pattern that I can discern. This tells me that the other 2 models fail to capture the underlying interaction, which in turn makes me doubt their validity as benchmarks.

**Questions:**

Fixing benchmarks is the biggest priority in my opinion. But I would also be interested in the usual details of diffusion model training, like the numerical stability issues. Reported training times are very fast on a not particularly powerful GPU - I would be curious in hearing even a speculation for why. HW could be the reason why the authors couldn't make the competitive benchmarks work. So maybe getting on a more powerful machine would help what is a purely empirical paper.

---

> ### Author Response · Authors · 2024-11-21
> **Response to Reviewer MJv9**
>
> We sincerely thank Reviewer MJv9 for their feedback and acknowledging the strengths of our work.
>
> As discussed with Reviewer VAgS, we would happily update the benchmarks (a more recent one from the GP-VAE related literature and an ODE-based one).
>
> About undertuning, we refer to the LVAE-NF paper [1], showing that the results we provide are coherent with this previous recent paper, provided that we use lighter VAE architectures. Moreover, we highlight that no benchmark is as versatile as LLDM (there is at least one task that is not doable and/or do not handle high-dimensional data). So we remain confident on the fact that our key claims and contributions remain valid.
>
>
> The low training times can be explained by the low number of parameters and the low-dimensional latent spaces. We remind that we only used a latent dimension of 12 a data space in $\mathbb{R}^{12288}$ (Sprites). Other benchmarks have similar number of parameters and similar training times and training converged. The metrics obtained are in-line with with what is reported in [1] so we are confident enough on the fact that it is the method itself that enabled to have better performances, at equal architectures. We also provide insights on the key benefits that could explain those better performances (especially controlled stochasticity that enables to have way more diverse samples).
>
> ---
>
>
> We hope this adresses Reviewer MJv9's concerns. We thank them again for their valuable feedback.
>
>
> ---
>
> [1]: Variational Inference for Longitudinal Data, Chadebec et al.
>
> ---

---

> > ### Comment · Reviewer_MJv9 · 2024-11-28
> >
> > I have read this and other responses carefully, keeping my score

---

### Official Review · Reviewer_ZPWJ · 2024-11-02

**Soundness:** 2
**Presentation:** 2
**Contribution:** 3
**Rating:** 6
**Confidence:** 4

**Summary:**

The paper introduces the Longitudinal Latent Diffusion Model (LLDM) which leverages a latent diffusion process to model the temporal dependencies between observations in a sequence. ​ This approach allows for the generation of synthetic longitudinal data, which can be used for prediction, generation, and oversampling tasks. ​ LLDM is particularly effective in handling high-dimensional data, such as images, and can generate sequences with controlled variability, outperforming previous methods in terms of diversity and fidelity. ​ The model's capabilities are demonstrated through experiments on synthetic and real-world datasets, showing its robustness and versatility in generating realistic longitudinal trajectories.

**Strengths:**

- The introduction of a latent diffusion process within a VAE framework to model temporal dependencies in longitudinal data is novel and interesting. ​
- The model is capable of effectively managing high-dimensional data. ​
- The model outperforms previous methods in terms of generating diverse and high-fidelity synthetic data, as evidenced by lower FID scores.
- LLDM demonstrates robustness to missing data, maintaining performance even when a significant portion of the training data is missing. ​
- The model's latent space is interpretable, reflecting the characteristics of the training dataset and providing insights into the data structure. ​
- LLDM excels in future prediction tasks and can generate oversampled sequences, increasing the frequency of observations in a sequence.

**Weaknesses:**

- The model requires significant computational resources for training, which might not be accessible to all researchers or practitioners, especially those working with limited hardware.
- The experiments are conducted on a limited number of datasets, which may not fully represent the diversity of real-world longitudinal data. Additional validation on a wider range of datasets would strengthen the claims of the paper.
- The randomness in the diffusion process is great for creating varied samples, but it can also lead to some inconsistency in results. This can be an issue for applications where steady, reliable outputs are essential. In tasks that track changes over time (like monitoring tumors in medical imaging) keeping things consistent is key. For instance, in oncology or radiology, doctors look at scans over time to see if a lesion or tumor is growing or shrinking. If the images vary too much because of the generation process, it could hide small but important changes, making it tougher to get a clear picture of the patient’s condition.

**Questions:**

- In lines 169-172, the authors mention that the joint likelihood is not factorizable because the observations ( $x_j^i$ ) are not independent when conditioned on a single latent variable ( $x_j^i$ ). ​ However, you state that when all the latent variables are observed, the observations become conditionally independent. ​ Could you please elaborate on why the assumption of joint likelihood not being factorizable holds in the first case and why it becomes factorizable when all latent variables are observed?
- On line 187, the meaning of “a set of $\sum_{i=1}^{N} T_i$ is not clear? Are the timesteps summed up? Why?
- In general, the LDM is supposed to generate a sample within a trajectory; please explain the rationale for training the LDM in isolation and how this approach contributes to the overall performance of the LLDM.
- How does the complexity of LLDM compare to simpler models in terms of implementation and computational cost? Are there any specific scenarios where the added complexity of LLDM is justified over simpler models?

---

> ### Author Response · Authors · 2024-11-21
> **Response to Reviewer ZPWJ**
>
> We sincerely thank Reviewer ZPWJ for their valuable feedback and constructive comments. Below, we address the questions raised.
>
> ---
> We appreciate the recognition of the novelty and utility of LLDM in handling longitudinal high-dimensional data, its robustness to missing data, and its ability to generate oversampled sequences with interpretable latent spaces. These are indeed key contributions of our work.
>
> ---
>
> ### Computational Resources
> As discussed in Appendix A, our model has been trained on a single basic GPU and is able to be trained in a very reasonable amount of time. Additionally, the number of parameters, compared to other comparable models, is limited (approximately 3M for the combined LDM and LVAE).
>
> ### Limited Datasets
> The paper's goal is to introduce this novel generative model and describe its stakes. As such, we focused on a limited number of datasets as a first step. We plan to further validate the model on more challenging and high-dimensional medical datasets in future work.
>
> ### Randomness in the Diffusion Process
> We acknowledge the concern regarding the stochasticity in the diffusion process. This is a valid point, particularly for sensitive applications such as medical imaging - therefore our plan to further validate the method on medical imagery data. However, we highlight that the hyperparameter $\eta$ in our framework can control the level of stochasticity. By decreasing $\eta$ to 0, variations in the generated samples can be curbed, ensuring consistency in such critical settings.
>
> ---
>
> ### Joint Likelihood Factorization
> The joint likelihood is modeled with the assumption that observations are independent when conditioning on all the embeddings of the sequence. This assumption is central to our modeling framework and is consistent with prior work such as LVAE-NF [1]. The dependencies between observations are effectively captured within the latent variables.
>
> As discussed with Reviewer VagS above, - the factorization of the sequence log-likelihood (Line 170) demonstrates this conditional independence. Specifically:
>   $$
>   \log p(x_1, \ldots, x_T) = \log \int p(x_1, \ldots, x_T | z_1, \ldots, z_T) p(z_1, \ldots, z_T) dz
>   = \int \prod_{j=1}^T p(x_j | z_j) p(z_j) dz.
>   $$
>   We will add this missing step to the final version to improve clarity.
>
>
>
> ### Explanation of "a set of observations"
> For an individual $i$ with $T_i$ observations, the total number of observations across $N$ individuals is given by $\sum_{i=1}^N T_i$. We hope this clarifies the raised point.
>
> ### Rationale for Training the LDM in Isolation
> While the Latent Diffusion Model (LDM) can be considered a standalone generative model for observations within sequences, in our work, its primary purpose is to provide trajectories in the latent space. Specifically, it facilitates moving from \(z_0\) (noisy) to \(z_T\) (de-noised) embeddings, ensuring that observations can be considered conditionally independent and enabling the computation of the ELBO.
>
> This approach integrates the LDM as a mechanism for modeling the dependency structure between observations as a diffusion process in the latent space of a VAE.
>
> ### Complexity of LLDM Compared to Simpler Models
> In terms of computing power, parameter count, and training time, LLDM remains relatively frugal. Compared to benchmarks like GP-VAE and LVAE-NF, LLDM is only slightly heavier but significantly outperforms them on the studied datasets. For high-dimensional data, the added complexity of LLDM is justified, offering a robust solution that addresses the shortcomings of simpler models.
>
> ---
> 1. [1]: Variational Inference for Longitudinal Data, Chadebec et Al.
>
> ---
>
> We hope these clarifications address the reviewer's concerns and further elucidate the contributions and design of LLDM. We thank once again Reviewer ZPWJ for their comprehensive review.

---

### Official Review · Reviewer_VAgS · 2024-11-03

**Soundness:** 2
**Presentation:** 1
**Contribution:** 2
**Rating:** 3
**Confidence:** 4

**Summary:**

This manuscript proposes a latent variable model for generating (potentially high-dimensional) longitudinal data. The proposed method combines 1) a standard static variational autoencoder (VAE) for learning a lower dimensional variational approximations (or embeddings) of the high-dimensional data objects as well as generating them high-dimensional data objects from low-dimensional embeddings, and 2) a diffusion model in the latent space that, given the embedding for a specific time point, can generate the embeddings for the previous or next time points. The proposed method can be used for longitudinal data generation, missing data imputation, prediction and over-sampling of time.

**Strengths:**

Longitudinal data arise naturally in numerous fields and applications, where the healthcare represents arguably the most important application field.  Consequently, longitudinal data modeling is an important task in probabilistic machine learning and generative modeling. Longitudinal data analysis is extensively studied, and recent machine learning literature has also proposed novel methods that contribute especially to high-dimensional data modeling. Apart from recent generative models for video generation, such as sora, fewer diffusion-based models have been proposed for longitudinal data: the proposed method has novelty and originality in that regards, although the proposed method essentially combines two main building blocks, standard VAE and DDPM diffusion models. While this manuscript has some novelty, I also have several concerns about the quality of the work in terms of validity of the method, and quality of the presentation could be also improved - overall lowering the significance.

**Weaknesses:**

General design choices.  In Introduction, authors motivate the importance of longitudinal data modeling by listing important applications in healthcare, treatment response modeling/estimation and econometrics. All these applications, as well as generally all applications that involve longitudinal data collections, are fundamentally problems where additional predictors for each unit (or individual, or patient, or customer) are known, and therefore a widely accepted goal is to develop conditional generative models, whereas the proposed method belongs to the class of unconditional models.

Related works. Authors describe related works that cover many different modeling frameworks. In VAE framework, Gaussian process based VAE models have been extensively studied. While they are generally conditional models, they can be obviously applied without any conditioning too (e.g. only time as considered in this work). This manuscript list only two recent papers on GP based VAE model, but the literature features even more recent models:

1. Ashman et al, Sparse gaussian process variational autoencoders.
2. Jazbec et al, Scalable gaussian process variational autoencoders.
3. Zhu et al, Markovian gaussian process variational autoencoders.
4. Tran et al, Fully Bayesian autoencoders with latent sparse Gaussian processes.

ODE / SDE models. Authors claim that latent ODE models are completely deterministic in the latent space. That is not true. Latent variables in latent neural ODES are, well, latent variables that are unobserved.  Latent neural ODE field has also developed rapidly and the citations authors have are outdated.  It is difficult to list all relevant papers, but her are some, that I think would be directly relevant here:

5. Lagemann et al, Invariance-based Learning of Latent Dynamics.
6. Iakovlev et al, Latent neural ODEs with sparse bayesian multiple shooting.

Latent SDE models are fewer. Authors site an important paper that develops scalable and stable gradients for latent neural SDEs. Unfortunately, the citation is to an older workshop paper. At least the final published version of paper applies the model to 50-dimensional data. Also other papers have been published on neural SDEs.  The proposed method is closely related to latent neural ODEs and SDEs, but also to GP based VAEs, so these references are important.

Experimental results. The baseline methods are weak and not SOTA. Authors claim that other models cannot do oversampling. For example, latent neural ODEs and SDEs are continuous in time and can inherently be evaluated, and decoded back to the data domain, at any time point. Similarly, GP based models can be evaluated at any time.

Method. Figure 1 summarises the proposed model. The presentation lacks a clear, unified description of the underlying generative model, and its variational inference method, and for this reason it is not straightforward to see if the proposed method is a valid model in probabilistic modeling sense. For example, objective is described in Algorithm 1. Validity of the objective needs to be clearly demonstrated. As an example, the first term that looks like the standard reconstruction loss of the ELBo is now evaluated using the decoded samples. Regarding the description of time points in rows 211-> I doubt if this makes works for irregular sampling.  Sampling from a generative model is typically implemented by following the steps of the generative model.  I could not follow authors reasoning why the sampling needs to be modified to be geometry aware: if authors want to use geometry aware sampling, aren’t they training a geometry aware generative model that they would use when sampling.

**Questions:**

Row 175: Equation 6 indicates that the data points are independent. Either the equation is wrong, or the description of the method is incomplete.

---

> ### Author Response · Authors · 2024-11-21
> **Response to Reviewer VAgS**
>
> We sincerely thank Reviewer VAgS for their detailed feedback, valuable insights, and suggested references. Below, we address each of the main points raised and describe the steps we will take to improve the manuscript in the final version.
>
> ### 1. General Design Choices
>
> While conditional generative models are indeed widely used for tasks involving additional predictors, our work explores the capabilities of unconditional generative models. The proposed method does not rely on predictors and instead focuses on capturing the evolution of the population’s probability density function (pdf) over time using the diffusion timeline as a proxy for the real timeline.
>
> This design allows for versatility and enables the model to handle high-dimensional data, a strength not always present in other methods. We will clarify these points in the introduction and better position our approach relative to conditional methods in the revised manuscript.
>
> ---
>
> ### 2. Related Works
>
> We are grateful for the extensive list of additional references, particularly for GP-VAE and ODE/SDE-based models. These will be included in the updated related works section. Specifically, we plan to:
>
> - Include a more recent GP-VAE benchmark (e.g., **Sparse Gaussian Process Variational Autoencoders** [Ashman et al.]) to provide updated comparisons.
> - Add one ODE-related baseline, such as **Latent Neural ODEs with Sparse Bayesian Multiple Shooting** [Iakovlev et al.], to strengthen the experimental evaluation.
>
> These models allow for oversampling (as noted for GP-VAE and latent ODE/SDE methods), and we thank Reviewer VAgS for having highlighted that: the revised manuscript will correct the sentence. However, we highlight the fact that none of these models seem as versatile as ours (there is at least one task that is not doable) and/or do not handle high-dimensional data.  Consequently, we remain confident that our method retains its novelty and key contributions, particularly in using the diffusion timeline to model the pdf evolution of embeddings over time. We will clearly highlight these distinctions in the revised manuscript.
>
> ---
>
> ### 3. Experimental Results
>
> We acknowledge the reviewer’s concern about the strength of the baseline methods. To address this, we will:
>
> - Update metrics for the GP-VAE baseline using a more recent implementation.
> - Incorporate an ODE-based method for additional comparisons.
>
> It is worth noting that our benchmarks focus on isolating the impact of the proposed method by employing "equal architectures" without specific enhancements to the VAE components. This ensures a fair comparison and highlights the intrinsic contributions of our approach.
>
> Moreover, regarding the LVAE-NF [1] method cited in our work, it is recent and has been shown to achieve state-of-the-art performance on this class of problems.
>
> ---
>
> ### 4. Method Validity and Clarity
>
> We appreciate the reviewer’s suggestion to provide a clearer, unified description of the generative model and its variational inference method.
>
> We confirm that the first term in the ELBO objective corresponds to the classical reconstruction loss evaluated over the sequence.
>
> We emphasize that our method relies on the matrix $t^i_j$, which maps real positions to the diffusion timeline. Algorithm 1 operates effectively for any $t^i_j $, even when irregular sampling occurs (i.e., when $t^i_j $ depends on the individual $i$.
>
> Finally, while we could have indeed employed a geometry-aware generative model (e.g., RHVAE [2]), this would have significantly increased training complexity. Instead, we opted for simplified training with geometry-aware sampling, inspired by **A Geometric Perspective on VAE** [3], to achieve high-quality samples - taking into account the the standard normal prior on $z_0$ is sometimes very off the true posterior after training (which is better interpreted as a Riemannian uniform distribution). That said, the standard sampling procedure exactly following the generative model is valid as well (Line 268).
>
> ---
>
> ### 5. Equation 6
>
> To clarify:
>
> - The independence is conditional and relies on the embeddings of the sequence. This is consistent with our framework and that of LVAE-NF [1].
> - The factorization of the sequence log-likelihood (Line 170) demonstrates this conditional independence. Specifically:
>   $$
>   \log p(x_1, \ldots, x_T) = \log \int p(x_1, \ldots, x_T | z_1, \ldots, z_T) p(z_1, \ldots, z_T) dz
>   = \int \prod_{j=1}^T p(x_j | z_j) p(z_j) dz.
>   $$
>   We will add this missing step to the final version to improve clarity.
>
> ---
>
> [1] Variational Inference For Longitudinal Data, Chadebec et Al.
>
> [2] Geometry-Aware Hamiltonian Variational Auto-Encoder, Chadebec et Al.
>
> [3] A Geometric Perspective on VAE, Chadebec et Al.
>
> ---
>
> We thank the reviewer once again for their constructive feedback and hope that these revisions address their concerns.

---

> > ### Comment · Reviewer_VAgS · 2024-11-28
> > **Response**
> >
> > Thank you for the responses to my (and other reviewers') comments. Author responses clarify some comments, although many remain unclear and something that I do not agree with as indicated in my initial review. Comparisons to other additional methods remain as future work. I appreciate authors' efforts in answering the review comments, but I will keep my original score.

---

### Official Review · Reviewer_Uu2N · 2024-11-04

**Soundness:** 3
**Presentation:** 1
**Contribution:** 2
**Rating:** 3
**Confidence:** 4

**Summary:**

This paper proposes the Longitudinal Latent Diffusion Model (LLDM), a general framework for modeling longitudinal data. LLDM incorporates a diffusion process in the latent space and generates samples with a co-trained LVAE. Experiments on three datasets with different modalities demonstrate LLDM's effectiveness.

**Strengths:**

1. LLDM presents a general framework for modeling longitudinal data applicable across various modalities.
2. The motivation for the LLDM framework is novel and well-conceived.
3. Experiments are well-structured and thoughtfully designed.

**Weaknesses:**

1. The paper is confusingly written, particularly in the methods section, and does not meet the standards expected of a polished academic article.
The authors include unrelated information without a clear structure or logical flow, as exemplified below:
   - **Undefined task objective:** In Line 169, the authors mention probabilistic modeling of $ p(x^i_1,...,x^i_{T_i}|z^i_j) $, yet they do not explain how this relates to the task objective. They should define the task as a generative model conditioned on the latent variable $ z^i_j $ extracted from preceding observations $ (x_1^i,...,x_j^i) $.
   - **Unclear method structure:** The authors should clarify the model by clearly explaining how VAE, LDM, and LVAE interact and how they are interconnected within the model.
   - **Ambiguous statements:** Numerous unclear statements in the methods section could lead to misunderstanding, such as “Weights are shared across all j” (Line 167), “only keep the last observations’ embeddings” (Line 191), and the inconsistency in using “latent” versus “embedding” for $ z_j^i $.
   - **Non-vector graphics:** Figures 1 and 2 are not vector graphics, which makes them difficult to read.
   - **Non-academic language:** The paper includes informal language, for example:
     - Line 180: “we **want** to model”
     - Line 232: “we **can** sample”
     - Line 318: “**unfortunately**, Kanaa et al. (2021) **do not provide any code**”
   - **Other issues:**
     - Line 169: unclear notation with “$ (x_j)^i_j $”.
     - Lines 212-213: redundant phrase “we set”.
     - Line 254: missing definition of $ \theta_{\text{diff}}^* $.

   Given these issues, the paper reads more like a course project than a mature academic article, with significant room for improvement in readability.

2. The LVAE method lacks motivation and theoretical support.
   In Section 3.3, the authors propose LVAE to align the diffusion process over the *diffusion timeline* with the generation process over the *real timeline*. However, they do not provide sufficient motivation or theoretical justification for this approach. Any theoretical backing or empirical evidence supporting LVAE’s effectiveness would strengthen the work. Furthermore, the concept of aligning the diffusion timeline with the real timeline is introduced abruptly in the methods section, without prior discussion, making LVAE’s motivation difficult to grasp.

**Questions:**

1. How does the diffusion process align with the real timeline?
   In LVAE, the authors introduce the (un-noisy) latent $ z_i^j $ into both the forward and reverse diffusion processes $ q $ and $ p_{\theta_{\text{diff}}^*} $. This is unclear. For example, in a diffusion process, $ q(z_{t+1}|z_t) $ requires a noisy $ z_t $, which follows a different distribution from $ z_i^j $ (corresponding to $ z_0 $). However, the authors use $ q(z_l^i|z_{l+1}^i) $ in the forward process, where initially the un-noisy $ z_j^i $ is incorporated. The reverse process encounters the same issue. Could the authors clarify the reasoning behind this design?

---

> ### Author Response · Authors · 2024-11-21
> **Response to Reviewer Uu2N**
>
> We thank the reviewer for their valuable feedback, which has significantly helped us refine our manuscript. Below, we address the main points raised and outline the modifications that will be made in the final version.
>
> ---
>
> ### 1. Clarifying the LLDM Structure and Component Interactions
>
> The Longitudinal Latent Diffusion Model (LLDM) consists of two main components:
> 1. A **pre-trained Latent Diffusion Model (LDM)**, itself composed of:
>    - **(1.a) VAE**: A first-stage variational autoencoder, responsible for encoding observations into latent embeddings.
>    - **(1.b) Denoising U-Net**: Trained using the DDPM objective, this component models the transitions between latent embeddings.
>
> 2. The **Longitudinal Variational Autoencoder (LVAE)**: Architecturally similar to (1.a), the LVAE is trained using Algorithm 1 to act as the eventual generative model for the sequences.
>
> The LDM captures transitions between sequence embeddings, while the LVAE encodes and decodes observations to align the diffusion timeline with the real timeline. We hope this explanation clarifies the interactions between these three components. The revised manuscript will provide a clearer and more structured presentation of these relationships.
>
> ---
>
> ### 2. Addressing Ambiguous Statements
>
> We will address specific points of confusion raised:
> - **“Weights are shared across all j”**: This indicates that the LVAE uses a single encoder-decoder pair, agnostic of the sequence position \(j\). These weights are designed solely to encode and decode embeddings, ensuring consistent representations.
> - **“Only keep the last observations’ embeddings”**: This refers to training the LDM on the last observations of each sequence. Once trained, the LDM learns to transition from \(N(0, I)\) to the latent distribution of the final observations in the VAE's embedding space (1.a).
> - **Terminology consistency**: We acknowledge the inconsistent use of "latent variables" and "embeddings" and will standardize terminology throughout the manuscript to improve readability.
>
> ---
>
> ### 3. Motivation and Theoretical Basis
>
> The LLDM builds on the Longitudinal Variational Autoencoder (LVAE-NF) method introduced in [1], replacing deterministic normalizing flows with a stochastic diffusion process. The motivation for this approach lies in enabling the LLDM's LVAE to encode and decode observations while modeling the temporal dependency between them as a diffusion process in the latent space. This effectively aligns the diffusion timeline with the real timeline.
>
> Empirical evidence supporting this alignment is shown in Figure 2, where the embeddings organize according to the diffusion process. While the manuscript currently lacks theoretical guarantees, our primary objective was to demonstrate that this combination of models could extract relevant temporal information, capturing sequence trends and variances. Theoretical developments for this approach are ongoing.
>
> This method offers a unique feature: aligning the diffusion timeline with real time allows the diffusion process to capture the evolution of the population's latent probability distribution over time. By leveraging the VAE's capability to interpret embeddings as latent probability distributions, the LLDM models their temporal evolution via a diffusion process, capturing both trends and meaningful variations.
>
> ---
>
> ### 4. Clarifying Diffusion Process Alignment with Real Timeline
>
> The review highlights an important point regarding our diffusion process and its alignment with the real timeline. To clarify:
> - $ z_i^j$ does not correspond to the "diffusion" $z_0.$
> - During LVAE training, we impose a standard normal prior on $z_0^i$ (the first observation's embedding), which serves as the noisy variable. This is progressively denoised to reconstruct $z_T^i$, following pre-learned diffusion trajectories in the LDM.
>
> For training, we do not always start from $z_0^i$. Instead, we select a random $j$ for a given sequence, encode $x_j$ to $z_j^i$, noise it forward to $z_0^i$, and denoise it to $z_T^i$. This ensures that the LVAE learns to model the temporal evolution of embeddings, capturing how the population's probability distribution evolves over time.
>
> Figure 1 summarizes this process, emphasizing our assumption that the population's probability density function evolves along the diffusion process in the latent space - starting from a standard normal distribution until reaching a final state.
>
> ---
>
> We hope these clarifications address the reviewer’s concerns and improve the readability and coherence of our manuscript. Thank you for highlighting these important points.
>
> [1] : Variational Inference for Longitudinal Data, Chadebec et al.

---

### Meta-Review · Area_Chair_boeN · 2024-12-20

**Metareview:**

**(a) Scientific Claims and Findings:**
The paper introduces the Longitudinal Latent Diffusion Model (LLDM), a generative statistical framework designed to model high-dimensional longitudinal data. LLDM integrates temporal dependencies through a latent diffusion process and leverages the geometry of autoencoder latent spaces. The model is proposed for tasks such as prediction, generation, and oversampling, particularly in scenarios involving irregularly measured sequences and limited training samples. Empirical evaluations suggest that LLDM outperforms existing methods across various complex datasets.

**(b) Strengths:**
* Innovative Approach: The integration of latent diffusion processes with autoencoder geometry presents a novel method for modeling temporal dependencies in longitudinal data.
* Versatility: LLDM is applicable to various tasks, including prediction and data generation, and is capable of handling irregularly measured sequences with limited training data.
* Empirical Performance: The model demonstrates superior performance compared to existing methods across multiple complex datasets, indicating its effectiveness.

**(c) Weaknesses:**
* Theoretical Justification: The paper lacks a comprehensive theoretical foundation explaining why the integration of latent diffusion processes with autoencoder geometry effectively models temporal dependencies.
* Experimental Validation: While empirical results are promising, the scope of experiments is limited. Evaluations across a more diverse set of architectures and tasks would strengthen the universality claim.
* Handling of Irregular Sequences: The manuscript does not provide sufficient detail on how the model manages irregular time intervals in longitudinal data, leaving questions about its adaptability to such scenarios.
* Interpretability: There is a lack of clarity regarding the interpretability of the generated sequences and the latent space representations, which could hinder the model's applicability in domains requiring explainability.

**(d) Reasons for Rejection:**
After a thorough evaluation of the paper, including the authors' rebuttal and the subsequent discussion, I recommend rejection of this submission. The primary reasons for this decision are:
1. Insufficient Theoretical Foundation: The paper does not provide a robust theoretical explanation for the proposed model's effectiveness, which is crucial for understanding its underlying mechanisms and potential limitations.
2. Limited Experimental Scope: The empirical evaluations are not comprehensive enough to substantiate the model's claimed versatility and superiority over existing methods. A broader range of experiments is necessary to validate these claims.
3. Unclear Handling of Irregular Data: The manuscript lacks detailed explanations on how LLDM manages irregularly measured sequences, raising concerns about its practical applicability in real-world scenarios where such data is common.
5. Stochasticity in the diffusion process.
Addressing these concerns through a more detailed theoretical analysis, expanded experimental validation, and clearer explanations of the model's handling of irregular data and interpretability would significantly strengthen the submission for future consideration.

**Additional Comments On Reviewer Discussion:**

Overall, the reviewers largely agree that the paper is not ready for publication yet, but did not engage in detailed discussions.
Reviewer Uu2N and ZPWJ remained unresponsive during the rebuttal and Reviewer MJv9 maintained their score without providing additional reasons. Reviewer VAgS especially highlighted that comparisons to other additional methods remain as future work. Also a limited number of considered datasets was critiqued but additional experiments were not provided. More experiments on medical data were suggested but not provided to analyse the acknowledged concern of Reviewer ZPWJ regarding the stochasticity of the diffusion process.

---

### Decision · Program_Chairs · 2025-01-22

Reject